# Homeostatic activation of aryl hydrocarbon receptor by dietary ligands dampens cutaneous allergic responses by controlling Langerhans cells migration

Adeline Cros[1], Alba De Juan[1], Renaud Leclère[2], Julio L Sampaio[3], Mabel San Roman[1], Mathieu Maurin[1], Sandrine Heurtebise-Chrétien[1], Elodie Segura[1]*

[1]Institut Curie, PSL Research University, INSERM, U932, Paris, France; [2]Institut Curie, PSL Research University, Plateforme de Pathologie Expérimentale, Paris, France; [3]Institut Curie, PSL Research University, Plateforme de Métabolomique et Lipidomique, Paris, France

*For correspondence: elodie.segura@curie.fr

**Abstract** Dietary compounds can affect the development of inflammatory responses at distant sites. However, the mechanisms involved remain incompletely understood. Here, we addressed the influence on allergic responses of dietary agonists of aryl hydrocarbon receptor (AhR). In cutaneous papain-induced allergy, we found that lack of dietary AhR ligands exacerbates allergic responses. This phenomenon was tissue-specific as airway allergy was unaffected by the diet. In addition, lack of dietary AhR ligands worsened asthma-like allergy in a model of 'atopic march.' Mice deprived of dietary AhR ligands displayed impaired Langerhans cell migration, leading to exaggerated T cell responses. Mechanistically, dietary AhR ligands regulated the inflammatory profile of epidermal cells, without affecting barrier function. In particular, we evidenced TGF-β hyperproduction in the skin of mice deprived of dietary AhR ligands, explaining Langerhans cell retention. Our work identifies an essential role for homeostatic activation of AhR by dietary ligands in the dampening of cutaneous allergic responses and uncovers the importance of the gut–skin axis in the development of allergic diseases.

## Editor's evaluation

This important study uncovers the role of Aryl Hydrocarbon Receptor (AhR) in tempering allergic responses. The authors present compelling data supporting the function of AhR ligands in limiting cutaneous allergic type 2 responses but not airway allergic responses, underscoring an interesting tissue-specific role of this axis. The work will be of broad interest to immunologists, including those with a special interest in mechanisms of regulation of allergy.

## Introduction

Development of immune-mediated diseases is affected by numerous environmental factors, including nutrition. Dietary compounds can modulate immune cells homeostasis and inflammatory immune responses (*Wu et al., 2018*), affecting in particular the susceptibility to allergy (*Julia et al., 2015*). However, the impact of individual nutrients and the molecular mechanisms involved remain incompletely understood. In particular, whether dietary metabolites that activate the aryl hydrocarbon receptor (AhR) influence type 2 allergic responses remains unclear.

Type 2 allergic responses result from dysregulated immune responses mediated mainly by IL4, IL5, and IL13. In the skin and lungs, allergens trigger the secretion by keratinocytes or epithelial cells of alarmins, such as thymic stromal lymphopoietin (TSLP) (*Deckers et al., 2017a*; *Akdis et al., 2020*). These soluble mediators stimulate the production of IL4, IL5, and IL13 by type 2 innate-like lymphoid cells (ILCs) and basophils, and the activation of dendritic cells (DCs) (*Deckers et al., 2017a*; *Akdis et al., 2020*). After their migration to the lymph nodes, activated DCs polarize CD4 T cells into Th2 cells that themselves secrete IL4, IL5, and IL13, thereby stimulating the production of IgE and amplifying the type 2 inflammatory response (*Deckers et al., 2017a*; *Akdis et al., 2020*). Nutritional compounds can affect multiple players of the type 2 inflammatory cascade through recognition by G-protein-coupled and nuclear receptors. For instance, mice fed on a high-fiber diet have increased levels of short-chain fatty acid propionate and decreased susceptibility to airway allergy due to impaired induction of Th2 polarization by lung DC (*Trompette et al., 2014*). Mice with a vitamin D-deficient diet display lung inflammation, high blood IgE levels, and increased numbers of lung Th2 cells (*Vasiliou et al., 2014*). Dietary interventions are therefore a promising strategy for preventing the development of allergic diseases, in particular in children (*Trambusti et al., 2020*; *Trikamjee et al., 2020*). However, a better understanding of the effects of each family of dietary compounds is needed.

AhR is a nuclear receptor-sensing metabolite produced mainly by the breakdown of food components or from tryptophan catabolism by intestinal microbiota (*Rothhammer and Quintana, 2019*; *De Juan and Segura, 2021*). Imbalance in gut-derived AhR ligands worsens inflammation in the intestine during inflammatory bowel disease (*Monteleone et al., 2011*; *Lamas et al., 2016*; *Hubbard et al., 2017*) and in the central nervous system during neuroinflammation (*Rothhammer et al., 2018*; *Rothhammer et al., 2016*). Whether dietary AhR ligands also play a role in other pathological contexts requires further investigation. In particular, AhR exerts broad functions in barrier tissues including skin and lung (*Esser and Rannug, 2015*), where AhR activation has been reported to limit inflammation (*Di Meglio et al., 2014*; *Beamer and Shepherd, 2013*). The impact of dietary AhR ligands in allergic responses at such barrier sites remains unknown.

In this study, we explored the role of dietary AhR ligands in the development of type 2 allergic responses using papain as a model allergen. We showed that lack of dietary AhR ligands exacerbates cutaneous, but not airway, papain-induced allergy. In addition, we found that lack of dietary AhR ligands during allergen cutaneous sensitization worsened asthma-like airway allergy in a recall phase, a model for 'atopic march.' We demonstrated that lack of dietary AhR ligands impacts the inflammatory profile of epidermal cells and increases the production of bioactive TGF-β, causing the retention of Langerhans cells in the skin, in turn leading to exaggerated Th2 responses in the lymph nodes. Our results identify a major role for dietary AhR ligands in the modulation of cutaneous allergic responses.

## Results

### Lack of dietary AhR ligands exacerbates cutaneous allergic type 2 responses

To study the impact of dietary AhR agonists on allergic responses, we sought to compare mice fed with diets either poor or rich in AhR agonists. Normal mouse chow contains phytochemicals that can act as precursors of AhR agonists, in particular indole-3-carbinole (I3C) (*Bjeldanes et al., 1991*), which is detected in the serum of mice fed on normal chow diet (*Figure 1—figure supplement 1A*). To avoid confounding effects, we used a standard purified diet (AIN-93M) that is naturally poor in phytochemicals (hereafter termed 'AhR-poor diet') and the same diet enriched for I3C (hereafter termed 'I3C diet'). As a proof of principle, we verified that I3C concentration in the serum of mice fed on the AhR-poor or I3C diet was significantly different (*Figure 1—figure supplement 1A*). To confirm that these diets induced different levels of AhR activation in vivo, we analyzed the expression in liver cells of the canonical AhR target gene *Cyp1a1*. Mice fed on the normal chow and I3C diets showed similar expression of *Cyp1a1*, while mice fed on the AhR-poor diet had almost undetectable expression of *Cyp1a1* (*Figure 1—figure supplement 1B*), validating that the AhR-poor diet contains negligeable amounts of natural AhR agonists. To study type 2 allergic responses, we chose the model protease allergen papain (*Shimura et al., 2016*; *Kamijo et al., 2013*; *Sokol et al., 2008*). To validate that the I3C diet would mimic normal conditions in this model, we analyzed Th2 cells induction in mice fed on normal chow or I3C diet. After footpad injection of papain, we observed similar Th2 induction in the

draining lymph nodes in both groups (*Figure 1—figure supplement 1C*). IL4 and IL13 were increased by papain exposure, while IL5 secretion was not significantly different from the control condition. Finally, to confirm that only AhR agonists from the diet were modulated in this setting, we measured the serum concentration of L-kynurenin (produced by host metabolism), 3,3'-diindolylmethane (DIM, generated by the degradation of I3C in the stomach), and indole-3-acetic acid (produced by microbiota metabolism from food components) (*De Juan and Segura, 2021*; *Bjeldanes et al., 1991*). DIM, but not L-kynurenin or indole-3-acetic acid, was decreased in the serum of mice fed on the AhR-poor diet (*Figure 1—figure supplement 1D*). Collectively, these results validate our experimental set up. For the rest of the study, we compared mice fed on the I3C and AhR-poor diets.

We first analyzed cutaneous allergic responses to papain challenge in the footpad. Histological analysis showed epidermal hyperplasia upon papain exposure only in the foot of mice fed on the AhR-poor diet (*Figure 1A and B* and *Figure 1—figure supplement 1E*). To assess the induction of Th2 responses, we analyzed draining lymph nodes after 6 d. The number of CD4 T cells and B cells was similar in mice fed on the IC3 or AhR-poor diets (*Figure 1C*), suggesting that the diet had no impact on CD4 T cells or B cells proliferation. We restimulated normalized numbers of lymph nodes T cells ex vivo and measured cytokine secretion. While IL4 secretion was induced similarly in both groups of mice, the production by T cells of Th2 cytokines IL5 and IL13 was exacerbated in mice fed on the AhR-poor diet (*Figure 1D*). To confirm these results in a setting of epicutaneous sensitization, we applied papain topically on the skin, without any abrasion, and restimulated T cells from the draining lymph node after 6 d (*Figure 1E*). Similarly, we found that mice fed on the AhR-poor diet showed higher secretion of IL5 and IL13 upon papain exposure. The secretion of IL10, IFN-γ, and IL17A upon papain challenge was also increased in mice fed on the AhR-poor diet in the footpad setting (*Figure 1—figure supplement 1F*), but not with topical application (*Figure 1—figure supplement 1G*). We concluded that lack of dietary AhR ligands amplifies allergic responses after cutaneous allergen exposure.

## Lack of dietary AhR ligands does not impact airway allergic type 2 inflammation

To address whether this phenomenon occurred with other exposure routes, we used a model of asthma-like airway allergy induced by repetitive intranasal exposure to papain. Papain challenge induced airway inflammation with infiltration of eosinophils, monocytes, and CD4 T cells as assessed in the bronchoalveolar space (*Figure 2A* and *Figure 2—figure supplement 1*). There was no difference in cell numbers between mice fed on the I3C or AhR-poor diets (*Figure 2A*). We could also detect IL13 and IL5 in the bronchoalveolar lavage, but cytokine secretion was not increased by the AhR-poor diet compared to I3C diet (*Figure 2B*). The production of Th2 cytokines by T cells from the pulmonary lymph nodes was also similar between diets (*Figure 2C*). We also assessed airway hyperreactivity by measuring in lung tissues the expression of the genes coding for mucus protein Mucin 5a (*Muc5ac*) (*Young et al., 2007*) and CLCA1 (*Gob5*), a molecule produced by goblet cells during hyperplasia (*Leverkoehne and Gruber, 2002*; *Nakanishi et al., 2001*). Papain challenge increased the expression of *Muc5ac* and *Gob5* in the lungs, to a similar extent in mice fed on I3C or AhR-poor diets (*Figure 2D*). Finally, we measured the concentration of IgE in the blood and found comparable levels in both groups of papain-exposed mice (*Figure 2E*). Collectively, these results indicate that the lack of dietary AhR ligands does not affect airway type 2 allergic responses.

## Lack of dietary AhR ligands worsens airway allergy after skin sensitization

Cutaneous allergen exposure can lead to airway allergy to the same allergen via the induction of Th2 memory CD4 T cells by skin DC (*Deckers et al., 2017a*), a phenomenon referred to as 'atopic march' (*Bantz et al., 2014*). Therefore, we addressed whether dietary AhR ligands impact asthma-like airway allergy after skin sensitization. To this aim, we fed mice on the I3C or AhR-poor diet only during the sensitization phase and placed all experimental groups on I3C diet 7 d after cutaneous exposure to papain or vehicle via footpad injection. Then, all groups were exposed to papain intranasally. While the number of alveolar macrophages was unaffected by the diet, lack of dietary AhR ligands during the sensitization phase increased the infiltration in the bronchoalveolar space of eosinophils, monocytes, and CD4 T cells (*Figure 3A*). IL13 secretion was higher in the bronchoalveolar lavage of mice fed on the AhR-poor diet during the sensitization phase, and IL5 concentration showed an increased

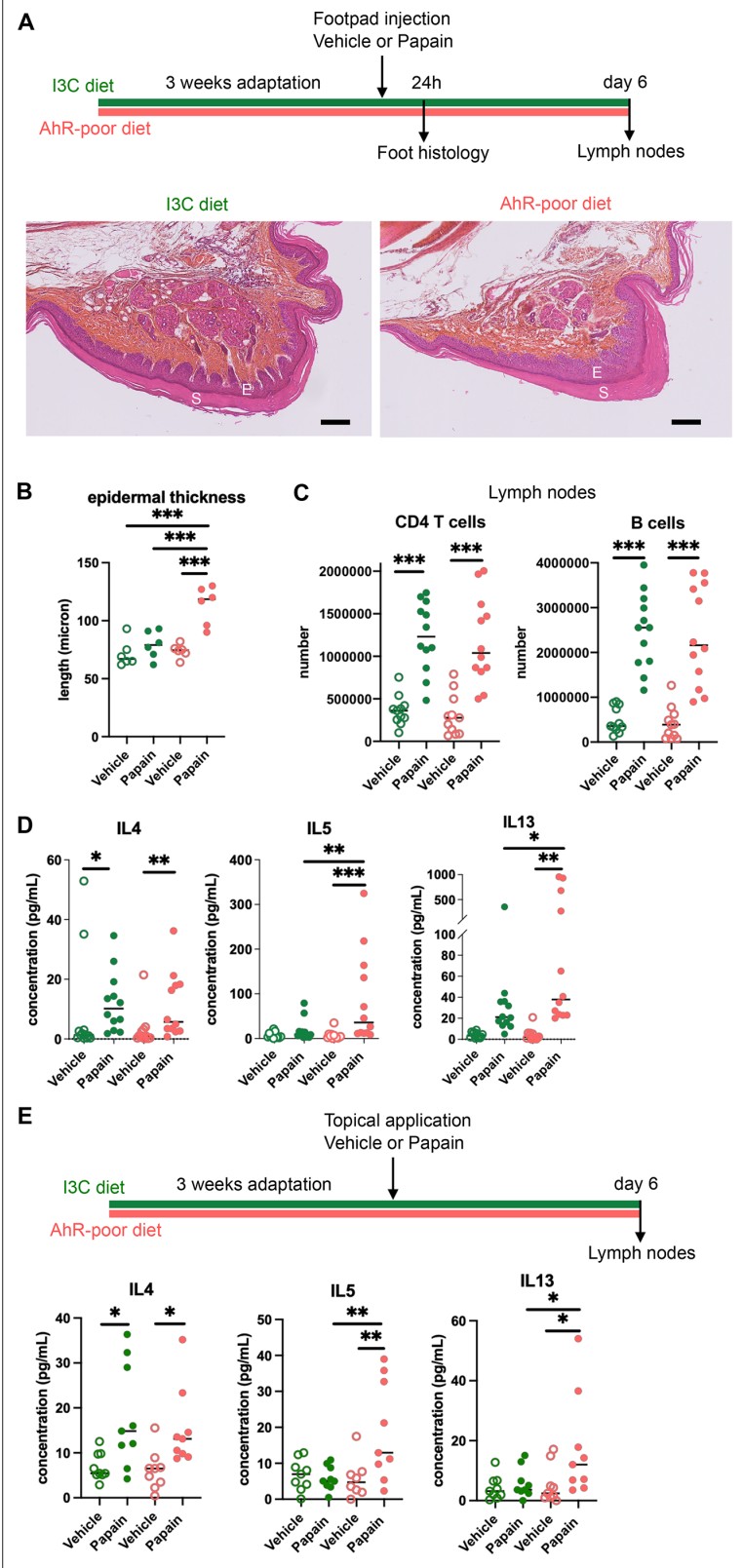

**Figure 1.** Lack of dietary aryl hydrocarbon receptor (AhR) ligands exacerbates cutaneous allergic type 2 responses. Mice were placed on AhR-poor diet or enriched in indole-3-carbinol (I3C diet) for 3 wk of adaptation prior to the start of experiments. (**A–D**) Papain or vehicle (PBS) was injected in the footpad at day 0. Mice were sacrificed for analysis either after 24 hr or at day 6. (**A, B**) Tissues were analyzed by histology 24 hr after papain injection

*Figure 1 continued on next page*

*Figure 1 continued*

(hematoxylin, eosin, and Safran staining). (**A**) Representative results (n = 6 per condition). E = epidermis, S = *stratum corneum*. Bar = 100 µm. (**B**) Epidermal thickness was measured on images. Median is shown (n = 6 in two independent experiments). One-way ANOVA. (**C, D**) After 6 d, cells from the draining lymph nodes were analyzed. (**C**) CD4 T cells and B cells counts. (**D**) Normalized numbers of T cells were restimulated ex vivo, and cytokine secretion was measured in the supernatant after 24 hr. Median is shown (n = 11–12 in three independent experiments). Kruskal–Wallis test. (**E**) Papain or vehicle (PBS) was applied topically at day 0. Mice were sacrificed for lymph nodes analysis at day 6. Normalized numbers of T cells were restimulated ex vivo, and cytokine secretion was measured in the supernatant after 24 hr. Median is shown (n = 9 in three independent experiments). Kruskal–Wallis test. For all panels *p<0.05; **p<0.01; ***p<0.001.

The online version of this article includes the following figure supplement(s) for figure 1:

**Figure supplement 1.** Comparable cutaneous allergic Th2 responses in mice fed with normal chow and indole-3-carbinol (I3C) diet.

tendency that was not statistically significant (*Figure 3B*). Finally, the expression of *Muc5ac* and of *Gob5* in lung tissues was significantly increased in mice deprived of dietary AhR ligands during the sensitization phase (*Figure 3C*). These results show that the lack of dietary AhR ligands during cutaneous allergen exposure worsens subsequent airway allergy.

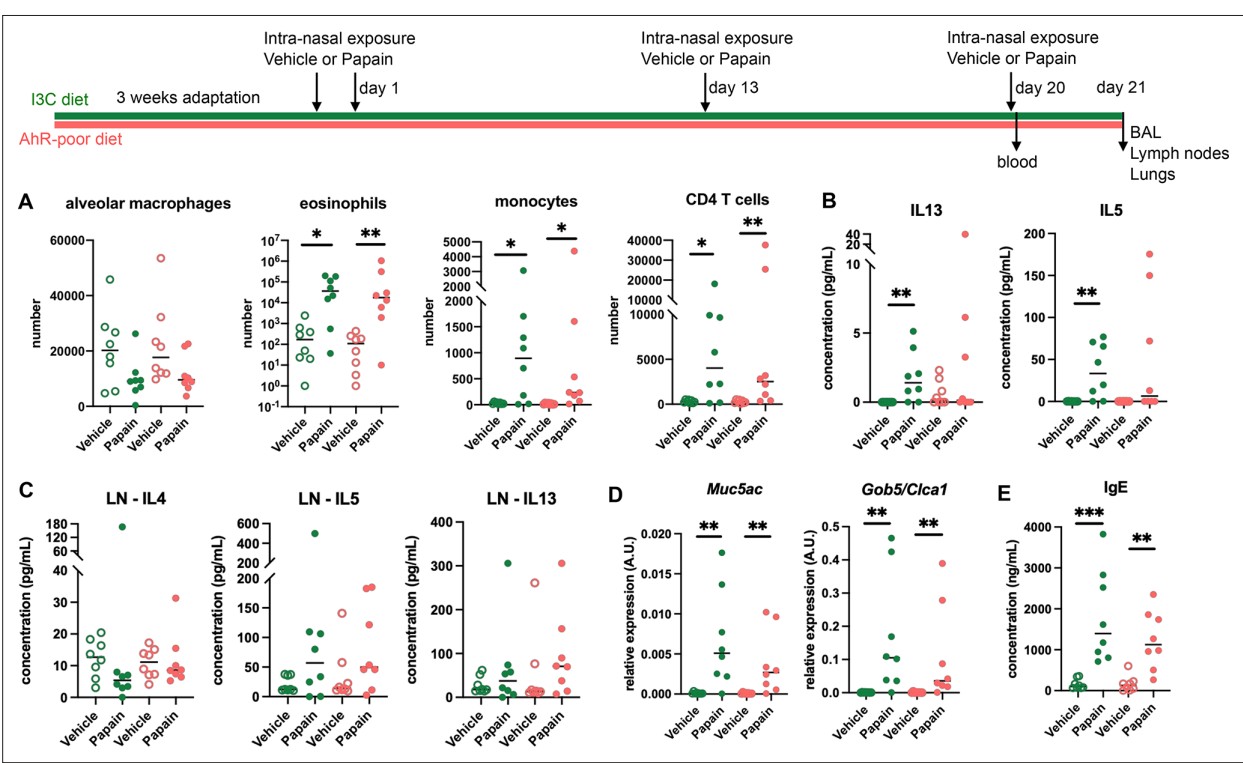

**Figure 2.** Lack of dietary aryl hydrocarbon receptor (AhR) ligands does not impact airway allergic type 2 inflammation. Mice were placed on AhR-poor or indole-3-carbinol (I3C) diet for 3 wk of adaptation prior to the start of experiments. Mice were exposed to papain or vehicle (PBS) intranasally four times at days 0, 1, 13, and 20. (**A–D**) 24 hr after the last exposure, tissues were analyzed. (**A**) Cell counts in bronchoalveolar lavage. (**B**) Cytokine concentration in bronchoalveolar lavage. (**C**) Normalized numbers of lymph nodes T cells were restimulated ex vivo, and cytokine secretion was measured in the supernatant after 24 hr. LN = lymph nodes. (**D**) Lung lysates were analyzed by RT-qPCR. (**E**) Blood was collected at day 20 and IgE concentration measured in the serum. For all panels, median is shown (n = 8 in two independent experiments). Kruskal–Wallis test. *p<0.05; **p<0.01; ***p<0.001.

The online version of this article includes the following figure supplement(s) for figure 2:

**Figure supplement 1.** Gating strategy for bronchoalveolar cells.

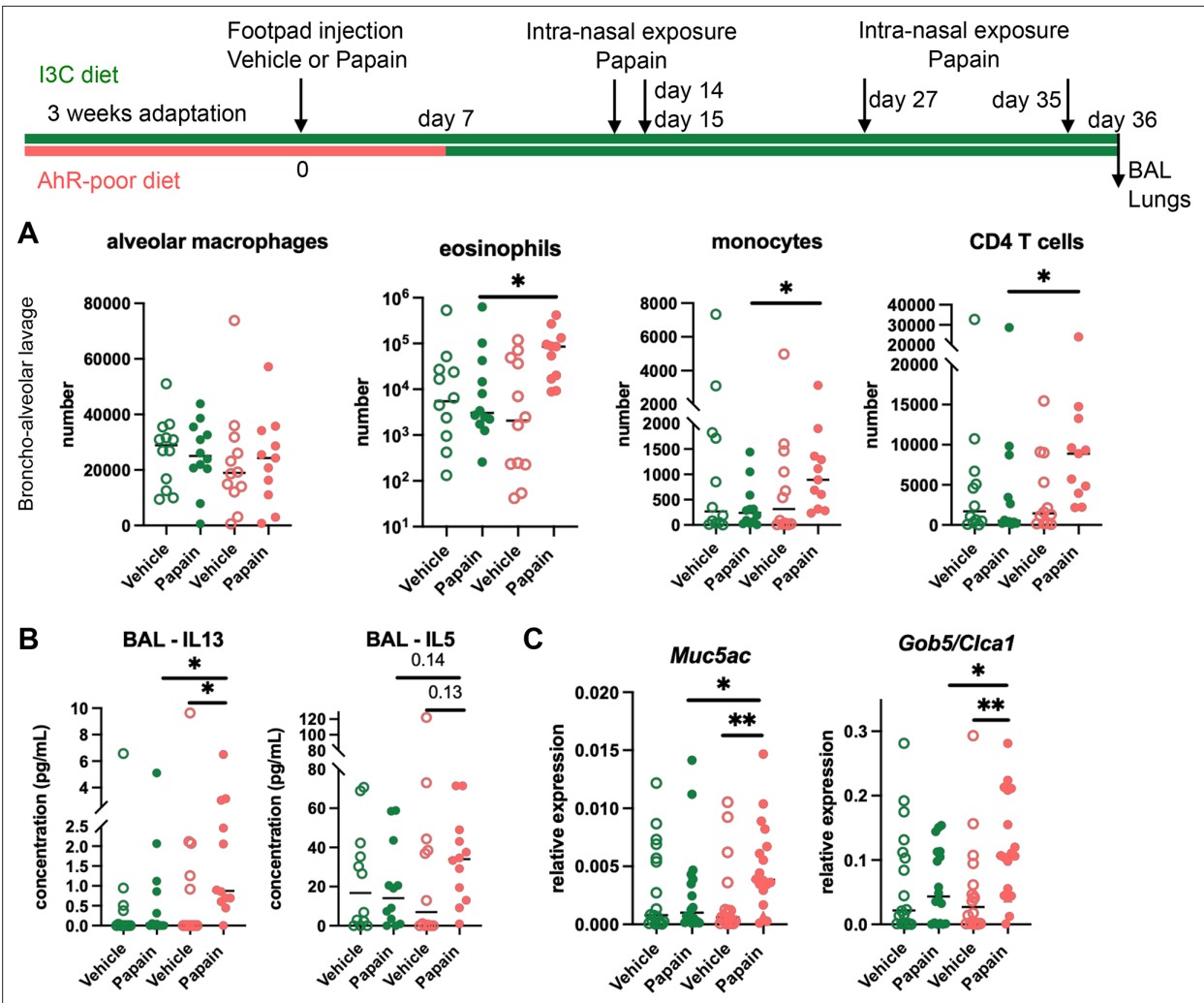

**Figure 3.** Lack of dietary aryl hydrocarbon receptor (AhR) ligands worsens airway allergy after skin sensitization. Mice were placed on AhR-poor or indole-3-carbinol (I3C) diet for 3 wk of adaptation prior to the start of experiments. Papain or vehicle (PBS) was injected in the footpad at day 0. At day 7, all mice were placed on the I3C diet. Mice were exposed to papain intranasally four times at days 14, 15, 27, and 35. 24 hr after the last exposure, bronchoalveolar lavage (BAL) and lung tissue were analyzed. (**A**) Cell counts in BAL (n = 12 in three independent experiments). (**B**) Cytokine concentration in BAL (n = 12 in three independent experiments). (**C**) Lung lysates were analyzed by RT-qPCR (n=18 in four independent experiments). For all panels, median is shown. Kruskal–Wallis test. *p<0.05; **p<0.01.

## Lack of dietary AhR ligands impairs Langerhans cells migration that increases Th2 responses in the lymph nodes

The contrasting results obtained in cutaneous versus airway papain-induced allergy suggest tissue-specific effects. Because of their central role in Th2 cells induction, we analyzed DC populations. One difference between skin and lung tissues is the presence of Langerhans cells, residing in the epidermis and absent from the lungs. AhR is expressed by Langerhans cells and has been proposed to regulate their numbers in the epidermis (**Hong et al., 2020**). To address whether lack of dietary AhR ligands affects Langerhans cells maintenance in the epidermis, we quantified epidermal Langerhans cells from mice fed on the I3C or AhR-poor diet. We first imaged the epidermis of mice expressing green fluorescent protein (GFP) under the promoter of *Cd207/Langerin* gene (**Kissenpfennig et al., 2005**; *Figure 4A*). We found no significant difference between diets in Langerhans cells density. To confirm this observation using another approach, we analyzed footpad epidermal cells by flow cytometry (*Figure 4—figure supplement 1A and B*) and found no significant difference between diets in Langerhans cells numbers. We then analyzed skin DC migration to the skin-draining lymph nodes. We assessed DC numbers 24 hr and 48 hr after papain footpad injection compared with

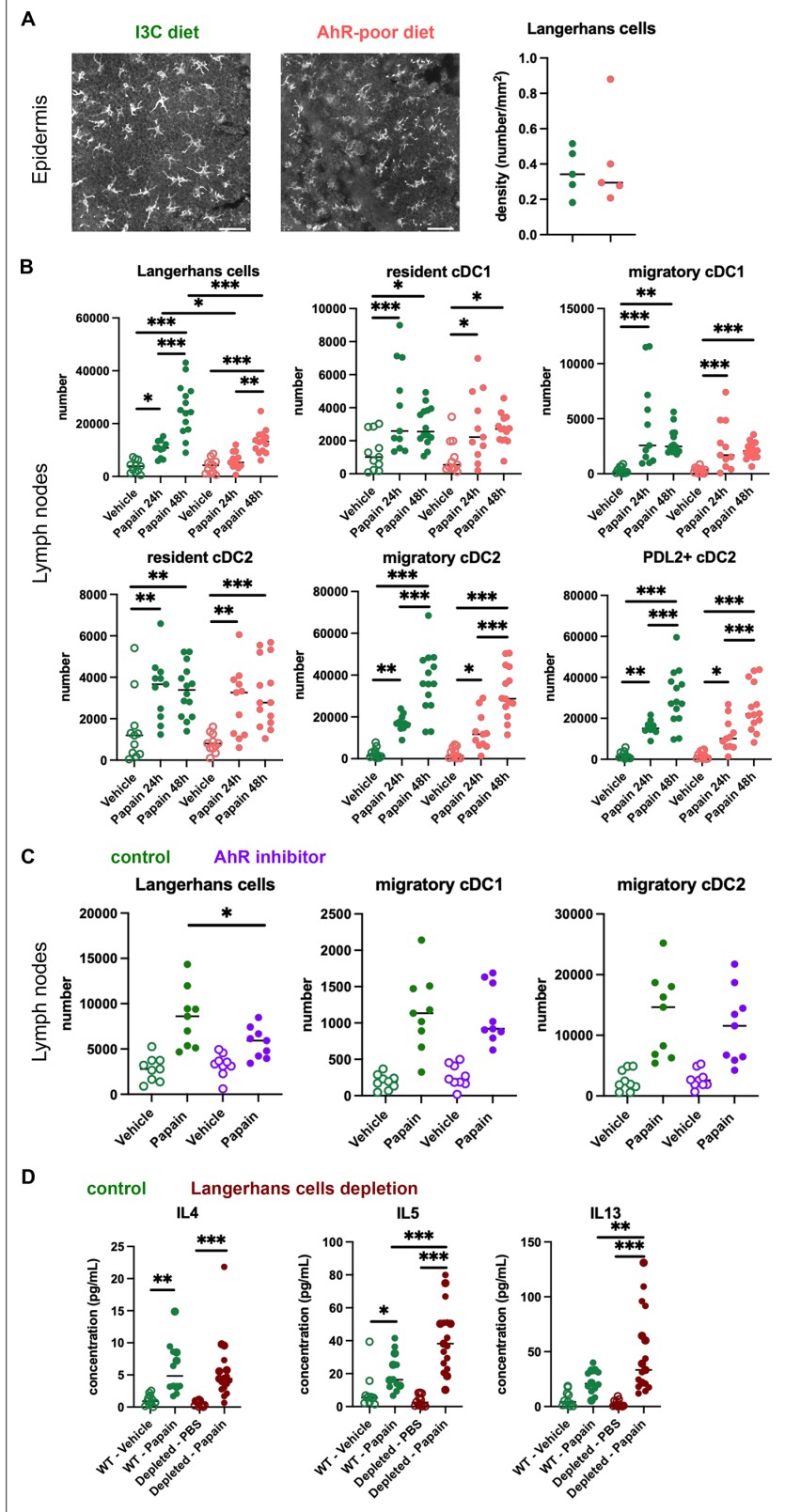

**Figure 4.** Lack of dietary aryl hydrocarbon receptor (AhR) ligands impairs Langerhans cells migration, thereby increasing Th2 responses. (**A, B**) Mice were placed on AhR-poor or indole-3-carbinol (I3C) diet for 3 wk of adaptation prior to the start of experiments. (**A**) Density of epidermal Langerhans cells was assessed by imaging *Cd207*(*Langerin*)-eGFP+ cells. Representative images are shown (n = 5 per condition). Median is shown.

*Figure 4 continued on next page*

*Figure 4 continued*

(**B–D**) Papain or vehicle (PBS) was injected in the footpad at day 0. (**B**) After 24 hr (vehicle and papain) or 48 hr (papain), dendritic cells numbers were assessed in the draining lymph nodes. Median is shown (n = 11–14 in three independent experiments). (**C, D**) Mice were fed with the I3C diet for 3 wk of adaptation prior to the start of experiments. (**C**) Mice were treated with vehicle or AhR inhibitor CH-223191 at day 0. After 48 hr, dendritic cells numbers were assessed in the draining lymph nodes. Median is shown (n = 9 in three independent experiments). (**D**) *Cd207*(*Langerin*)-DTR mice or WT littermates were injected with Diphtheria Toxin 3 d prior to papain treatment. At day 6 after papain treatment, cells from the draining lymph nodes were analyzed. Normalized numbers of lymph nodes T cells were restimulated ex vivo, and cytokine secretion was measured in the supernatant after 24 hr. Median is shown (n = 11–17 in three independent experiments). One-way ANOVA. For all panels *p<0.05; **p<0.01; ***p<0.001.

The online version of this article includes the following figure supplement(s) for figure 4:

**Figure supplement 1.** Analysis of cutaneous allergy.

---

vehicle (*Figure 4—figure supplement 1C and D*). Langerhans cells numbers in the lymph nodes were significantly lower upon papain exposure in mice fed on the AhR-poor diet (*Figure 4B*). By contrast, papain exposure increased the number of migratory skin cDC1 and cDC2 independently of the diet (*Figure 4B*). In particular, migratory PDL2$^+$ cDC2 play an essential role in the induction of allergic Th2 responses (*Gao et al., 2013*; *Kumamoto et al., 2013*), and their number was similar in papain-treated mice from both diet groups (*Figure 4B*). In addition, the number of lymph node resident DC was similar in both diet groups (*Figure 4B*). The expression of MHC class II molecules or co-stimulatory molecule CD40 was not affected by the diet (*Figure 4—figure supplement 1E*). To confirm these results in the epicutaneous sensitization model, we analyzed draining lymph nodes 48 hr after papain topical application (*Figure 4—figure supplement 1F*). The numbers of resident and migratory cDC1 and cDC2 were increased in both diet groups upon papain exposure compared to vehicle treatment, while Langerhans cells number was significantly reduced in papain-treated mice fed on the AhR-poor diet. Collectively, our results show that lack of dietary AhR ligands impairs Langerhans cells migration upon papain challenge.

To confirm the role of AhR in this phenomenon, we examined DC migration after footpad papain challenge upon pharmacological inhibition of AhR. We used CH-223191, a potent AhR antagonist (*Zhao et al., 2010*), to treat mice fed on the I3C diet. Langerhans cells migration, but not that of cDC1 or cDC2, was significantly reduced in mice treated with the AhR inhibitor (*Figure 4C*). This result suggests that impaired Langerhans cells migration in mice fed with the AhR-poor diet is due to reduced AhR activation.

Langerhans cells play an essential role in dampening T cell responses in skin-draining lymph nodes (*Igyarto et al., 2009*; *Kaplan et al., 2005*; *Gomez de Agüero et al., 2012*). We hypothesized that reduced Langerhans cells presence in the lymph nodes could cause the observed exacerbated Th2 responses. To address this, we used *Cd207(Langerin)*-DTR mice, in which Langerhans cells can be depleted by Diphtheria Toxin injection (*Kissenpfennig et al., 2005*). We confirmed efficient Langerhans cells depletion in the footpad epidermis (*Figure 4—figure supplement 1G*). We injected papain or vehicle in the footpad of mice depleted of Langerhans cells or WT littermates (treated similarly with Diphtheria Toxin). To assess Th2 cells induction, we analyzed draining lymph nodes after 6 d. While IL4 secretion was similar in both groups, IL5 and IL13 secretion was significantly increased by Langerhans cells depletion (*Figure 4D*). These results mirror the observations made in mice fed with the AhR-poor diet (*Figure 1D*). We concluded that exacerbated Th2 responses in the absence of dietary AhR ligands are caused by impaired Langerhans cells migration to the lymph nodes.

## Dietary AhR ligands regulate the inflammatory profile of epidermal cells

To address how dietary AhR ligands control Langerhans cells migration, we analyzed the transcriptomic profile of epidermal cells in mice fed on the I3C or AhR-poor diet, in basal conditions (vehicle treatment) or upon allergen challenge (6 hr after papain treatment). Papain exposure induced a common transcriptomic program (*Figure 5—figure supplement 1A*), enriched for type I interferon pathway, cytokine-mediated signaling, matrix remodeling, and oncostatin M pathway (a regulator of keratinocyte activation) (*Boniface et al., 2007*; *Figure 5—figure supplement 1B*). Consistent with

pathway enrichment results, both papain-treated groups had significantly increased expression of inflammatory mediators genes (such as *Csf1*, *Ccl20*, and *Tslp*), interferon-stimulated genes (including *Stat1* and *Oasl2*) (*Figure 5—figure supplement 1C*), and genes involved in tissue repair and remodeling (such as *Areg*, *Il24*, *Mmp13*, *Osmr*, *Tgfa*, and *Vegfa*) (*Figure 5—figure supplement 1D*).

Differentially expressed genes between diets were overexpressed mostly in the epidermis of mice fed on the AhR-poor diet (*Figure 5A*), both in basal conditions and upon papain challenge (*Figure 5—figure supplement 1E*). Diet-modulated genes were enriched for regulation of extracellular matrix, cytokine signaling, and inflammatory responses, including the leptin pathway (*Figure 5B and C*). Consistent with this, mice fed on the AhR-poor diet had in papain-treated epidermis significantly higher expression of matrix components (such as *Col1a1*, *Col4a1*, *Col5a1*) and cell adhesion molecules (including *Itgb2* and *Itgb7*) (*Figure 5—figure supplement 1F*), and inflammatory genes including cytokine *Il1b* (*Figure 5D*) and chemokines *Cxcl1*, *Cxcl3*, *Cxcl5*, *Ccl2*, and *Ccl3* (*Figure 5D*). To confirm these results at the protein level, we analyzed the secretion of chemokines using skin explants. CCL2, CCL3, and CXCL1 were more released upon papain treatment in the skin of mice deprived of dietary AhR ligands (*Figure 5E*), consistent with mRNA expression. Collectively, these results show that homeostatic activation of AhR via dietary ligands down-modulates inflammatory pathways in epidermal cells.

## Diet-derived AhR ligands do not affect keratinocyte barrier

AhR is involved in keratinocyte differentiation and maintenance of skin barrier integrity (*Haas et al., 2016*; *van den Bogaard et al., 2015*). AhR ligands produced by commensal microbiota have been shown to be essential in this process (*Uberoi et al., 2021*). To address whether microbiota-derived and diet-derived AhR agonists have similar impact on epidermal genes, we reanalyzed public transcriptomic data from the epidermis of germ-free and specific pathogen-free (SPF) mice (*Uberoi et al., 2021*). The GO signature for keratinocyte differentiation was enriched in the epidermis of SPF mice (*Figure 5—figure supplement 2A*), consistent with previous findings that microbiota-derived AhR ligands control the expression of genes involved in barrier function such as *Cdsn*, *Cldn1*, *Dsc1*, *Dsg1a*, *Flg*, *Ivl*, and *Tjp3*, and of keratine molecules including *Krt10* (*Uberoi et al., 2021*; *Figure 5—figure supplement 2A*). By contrast, the genes overexpressed in the epidermis of mice fed on the AhR-poor diet were not enriched in any group (*Figure 5—figure supplement 2B*). In particular, the expression of chemokine genes or *Itgb8* was comparable between SPF and germ-free mice (*Figure 5—figure supplement 2B*). In addition, we found that lack of dietary AhR ligands had no impact on, or even increased, the expression of keratinocyte barrier genes (*Figure 5—figure supplement 2C*). Consistent with this, we did not observe by histology any diet-dependent defect in *stratum corneum*, the outer layer of epidermis formed by cornified keratinocytes (*Figure 1—figure supplement 1E* and *Figure 5—figure supplement 3A*). In addition, the ultra-structure of the epidermis was similar in both diet groups (*Figure 5—figure supplement 3B*). These observations suggest that the epidermal barrier function was not compromised in mice deprived of dietary AhR ligands and that diet-derived AhR ligands do not impact keratinocyte barrier.

## AhR activation regulates keratinocyte production of TGF-β in mouse and human

Langerhans cells migration is regulated by Tgf-β, which retains Langerhans cells in the epidermis (*Kel et al., 2010*; *Bobr et al., 2012*; *Mohammed et al., 2016*). Bioactive Tgf-β can be produced from the latent form upon cleavage by integrin-β8 expressed on keratinocytes (*Mohammed et al., 2016*; *Cambier et al., 2005*). One of the pathways enriched in the epidermis of mice fed on the AhR-poor diet was related to Tgf-β regulation (*Figure 5B*). Indeed, expression of *Itgb8* was significantly higher in the epidermis of mice fed on the AhR-poor diet, with or without papain treatment (*Figure 6A*), suggesting increased release of bioactive Tgf-β in the skin of mice deprived of dietary AhR ligands. To directly test this, we measured total and bioactive Tgf-β release in skin explants. The concentration of bioactive Tgf-β, but not that of total Tgf-β, was significantly higher in the skin of mice fed on the AhR-poor diet (*Figure 6B*).

To address the relevance of these results in human, we first reanalyzed public transcriptomic data from human skin exposed ex vivo to an AhR antagonist (Stemregenin-1, SR1) or agonist (6-formylindolo(3,2-b)carbazole, FICZ) (*Di Meglio et al., 2014*). FICZ is an endogenous AhR ligand

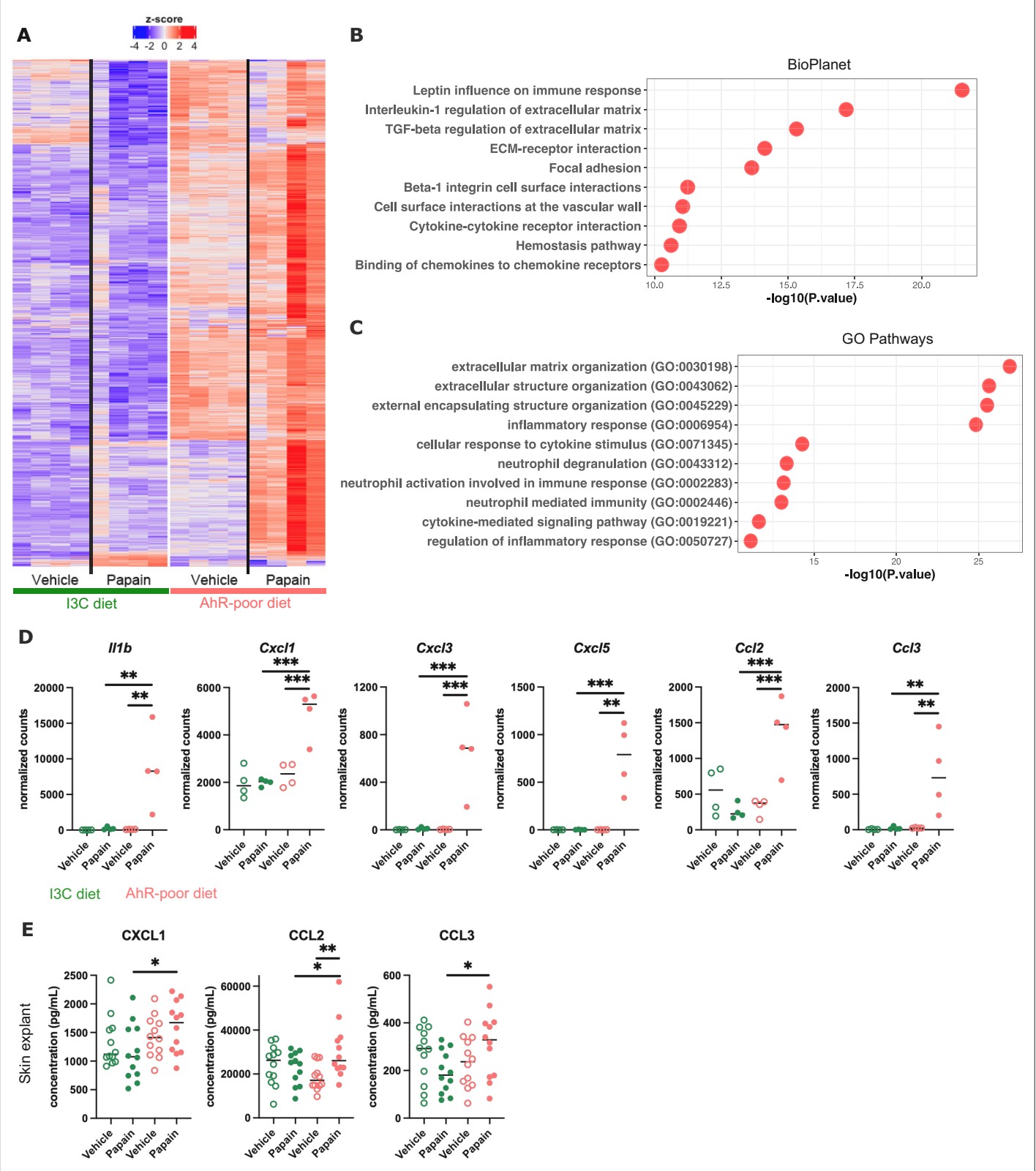

**Figure 5.** Dietary aryl hydrocarbon receptor (AhR) ligands regulate the inflammatory profile of epidermal cells. Mice were placed on AhR-poor or indole-3-carbinol (I3C) diet for 3 wk of adaptation prior to the start of experiments. Papain or vehicle (PBS) was injected in the footpad at day 0. (A–D) After 6 hr, epidermal cells were extracted and subjected to RNA-seq analysis. n = 4 biological replicates. (A) Diet-dependent differentially expressed genes. (B, C) Enrichment for biological pathways using BioPlanet (B) or Gene Ontology Biological Process databases (C). (D) Normalized counts for selected

*Figure 5 continued on next page*

*Figure 5 continued*

genes, median is shown. One-way ANOVA. (**E**) After 24 hr, footpad skin was collected and explants were cultured for 24 hr to prepare conditioned medium. Chemokine concentrations were measured in the medium. Median is shown (n = 12 in three independent experiments). Kruskal–Wallis test. For all panels *p<0.05; **p<0.01; ***p<0.001.

The online version of this article includes the following source data and figure supplement(s) for figure 5:

**Source data 1.** List of differentially expressed genes between conditions.

**Figure supplement 1.** Dietary aryl hydrocarbon receptor (AhR) ligands modulate the transcriptomic profile of epidermal cells.

**Figure supplement 2.** Diet-derived aryl hydrocarbon receptor (AhR) ligands do not impact keratinocyte barrier genes.

**Figure supplement 3.** Diet-derived aryl hydrocarbon receptor (AhR) ligands do not impact the structure of the skin barrier.

produced from the photo-oxidation of tryptophan (*Oberg et al., 2005*; *Wincent et al., 2009*). As expected, canonical AhR target genes were upregulated in the FICZ-treated samples (*Figure 6C*). By contrast, chemokine genes (*CCL2, CCL3, CXCL1, CXCL3, CXCL5*) and *ITGB8* were more expressed in SR1-treated skin (*Figure 6C*). These results are consistent with our transcriptomic data from mouse epidermis (*Figure 5D* and *Figure 6A*). To confirm that AhR activation regulates bioactive Tgf-β release by human keratinocytes, we used a human keratinocyte cell line (HaCaT cells). We cultured differentiated human keratinocytes in the presence of an AhR antagonist (SR1) or two different physiological AhR agonists, FICZ (produced by photo-oxidation) or DIM (produced from phytonutrients). AhR activation increased the expression of canonical target genes *CYP1A1* and *CYP1B1*, but decreased the expression of *ITGB8* (*Figure 6D*). To directly address the impact of AhR activation on Tgf-β secretion, we measured total and bioactive Tgf-β in the supernatant. While total Tgf-β concentration was similar between treatments, bioactive Tgf-β was significantly more released by SR1-treated keratinocytes (*Figure 6E*). Taken together, these results indicate that lack of AhR activation increases the production of bioactive Tgf-β in the skin in both mouse and human, thereby inhibiting Langerhans cells migration.

## Discussion

In this work, we showed that lack of dietary AhR ligands exacerbates cutaneous allergic Th2 responses and airway allergy after skin sensitization. We found that homeostatic activation of AhR by diet-derived agonists down-modulates inflammation pathways in epidermal cells. In particular, mice deprived of dietary AhR ligands displayed hyperproduction of bioactive Tgf-β in the skin, impairing Langerhans cells migration and their action as down-modulators of Th2 cells induction in the lymph nodes.

Previous work has reported a role for Langerhans cells in suppressing T cell responses in various models. In hapten-induced contact hypersensitivity, which is mediated by CD8 T cells, Langerhans cells are essential for tolerance to haptens (*Kaplan et al., 2005*), and to down-modulate T cell responses by producing IL10 (*Igyarto et al., 2009*) and by inducing regulatory CD4 T cells (*Gomez de Agüero et al., 2012*). In models of cutaneous sensitization to ovalbumin, Langerhans cells depletion increased Th2 cutaneous responses (*Marschall et al., 2021*; *Luo et al., 2019*) and T follicular helper cells (*Marschall et al., 2021*), as well as airway inflammation after intranasal recall (*Marschall et al., 2021*). Langerhans cells depletion also exacerbated lung Th2 responses after epicutaneous sensitization to house dust mite (*Deckers et al., 2017b*). It has been proposed that Langerhans cells produce IL10 upon exposure to allergens (*Luo et al., 2019*), but the mechanisms by which Langerhans cells regulate Th2 responses, and whether regulatory T cells are involved, remain to be better characterized. Consistent with these observations, we propose a model whereby dietary AhR ligands regulate the severity of cutaneous Th2 responses via the control of Langerhans cells migration, with decreased numbers of Langerhans cells in skin-draining lymph nodes leading to exacerbated Th2 cytokine production.

We found that reduced Langerhans cells numbers in lymph nodes increased T cell secretion of IL5 and IL13, but not that of IL4. Similar observations were made in a model of house dust mite allergy (*Deckers et al., 2017b*). These results are consistent with previous work showing that IL4 and IL13 are produced in lymph nodes by distinct populations of T cells during helminth infection (*Liang et al., 2012*), and that IL4 expression is regulated in vivo by distinct transcriptional mechanisms from IL5 and IL13 expression (*Kim et al., 1999*; *Tanaka et al., 2011*; *Bao and Reinhardt, 2015*). We speculate that Langerhans cells modulate IL5- and IL13-producing T cells, but not IL4-producing T cells.

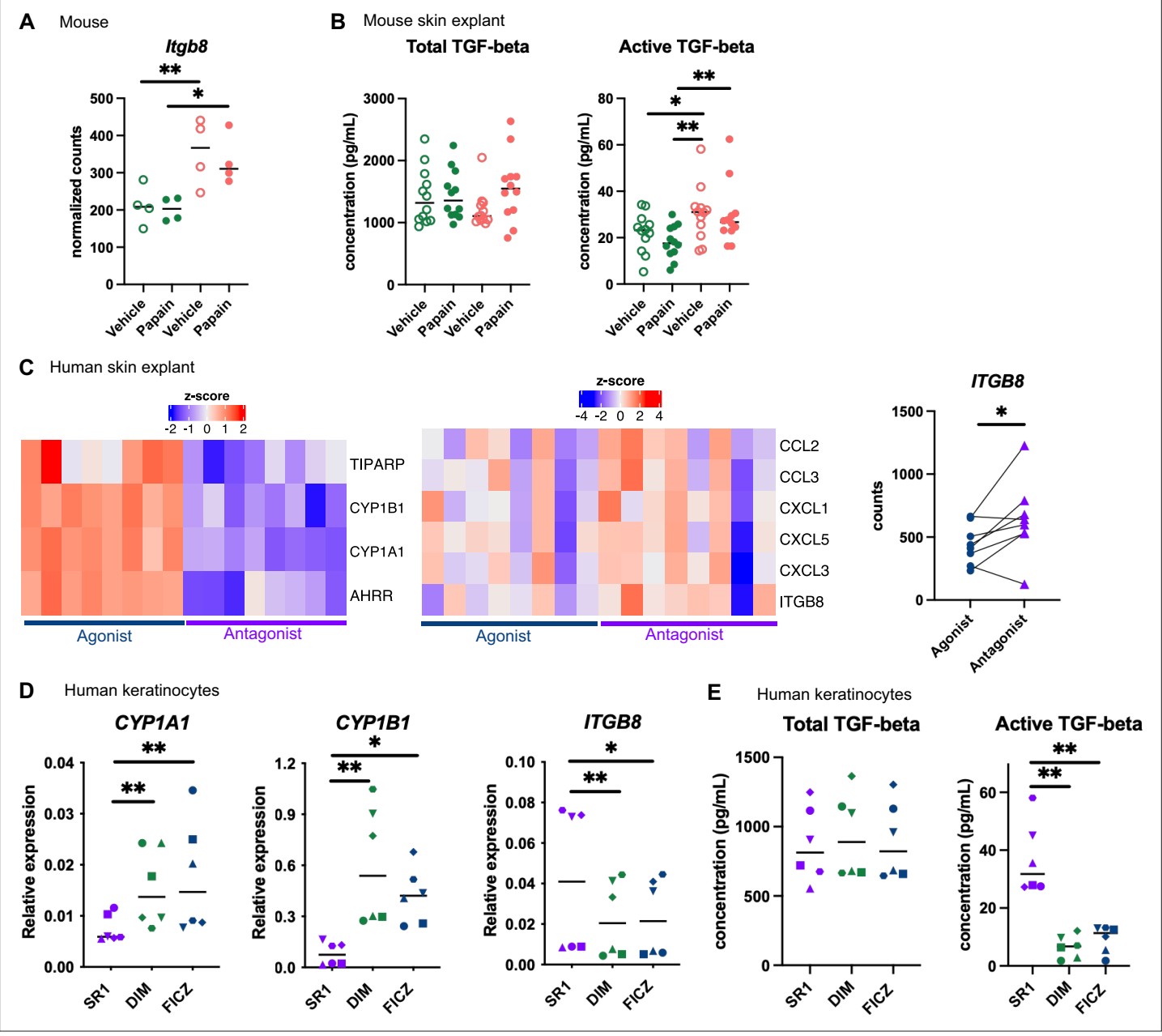

**Figure 6.** Aryl hydrocarbon receptor (AhR) activation regulates keratinocyte production of TGF-β in mouse and human. (**A, B**) Mice were placed on AhR-poor or indole-3-carbinol (I3C) diet for 3 wk of adaptation prior to the start of experiments. Papain or vehicle (PBS) was injected in the footpad at day 0. (**A**) After 6 hr, epidermal cells were extracted and subjected to RNA-seq analysis. n = 4 biological replicates. Normalized counts, median is shown. One-way ANOVA. (**B**) After 24 hr, footpad skin was collected and explants were cultured for 24 hr to prepare conditioned medium. TGF-β concentration was measured in the medium. Median is shown (n = 12 in three independent experiments). Kruskal–Wallis test. (**C**) RNA-seq data of human skin explant cultured in the presence of AhR agonist (FICZ) or AhR antagonist (SR1) was extracted from public source (GSE47944) (n = 8). Scaled expression of selected genes. Normalized counts for *ITGB8*, paired *t*-test. (**D, E**) Human HaCaT keratinocytes were cultured for 24 hr in the presence of AhR antagonist (SR1) or AhR agonists (DIM or FICZ). (**D**) Expression of selected genes was measured by RT-qPCR. Median is shown (n = 6 in three independent experiments, individual symbols represent paired conditions). Friedman test. (**E**) TGF-β concentration was measured in the supernatant. Median is shown (n = 6 in three independent experiments, individual symbols represent paired conditions). Friedman test. For all panels *p<0.05; **p<0.01; ***p<0.001.

Previous studies have proposed a role for AhR in regulating airway inflammation. In a model of ovalbumin-induced airway allergy, AhR-deficient mice developed more severe allergic responses due to the increased ability of AhR-deficient T cells to proliferate and the higher activation of AhR-deficient lung DC (*Thatcher et al., 2016*). In another study using ovalbumin-induced airway allergy, systemic injection of high doses of the AhR agonist FICZ reduced eosinophilia and Th2 cytokines in the lung and blood IgE levels (*Jeong et al., 2012*). In addition, in a model of cockroach allergen-induced allergy, mice deficient for AhR in type II alveolar epithelial cells had increased airway hyperreactivity including eosinophilia and elevated Th2 cytokine secretion due to dysregulated autophagy (*Wang et al., 2022*). By contrast, we found that airway allergic responses induced by papain exposure were not affected by the lack of dietary AhR ligands. Our results suggests that dietary AhR ligands do not play a role in the induction by lung DC of Th2 polarization per se, in cytokine secretion by lung ILC2 or in the inflammatory response of lung epithelial cells. This discrepancy could be explained by ligand-specific effects or by redundant effects of AhR agonists from different sources, the lack of dietary AhR ligand being compensated in the lung by endogenous or microbiota-derived ligands.

AhR can be activated by xenobiotics, diet-derived molecules, metabolites produced by microbiota metabolism, as well as endogenous ligands produced by cellular metabolism (*Rothhammer and Quintana, 2019*). Here, we specifically focused on natural diet-derived AhR agonists, delivered by a physiological route, that is, food absorption. In epidermal cells, we found that dietary AhR agonists regulate inflammatory pathways, but not keratinocyte barrier genes. By contrast, AhR ligands produced by skin commensal microbiota have been shown to control genes involved in keratinocyte differentiation and function (*Uberoi et al., 2021*). These results could be explained by ligand-specific effects. In a model of MC903-induced atopic dermatitis, topical application or oral administration of microbiota-derived AhR ligand indole-3-aldehyde (IAId) reduced skin inflammation and type 2 responses, which was dependent on AhR (*Yu et al., 2019*). However, application of other AhR agonists such as kynurenin (an endogenous ligand) or indole-3-acetic acid (produced by microbiota) had no impact on disease symptoms (*Yu et al., 2019*). Mechanistically, it was shown that AhR activation by IAId regulated TSLP production by keratinocytes (*Yu et al., 2019*). By contrast, here we did not find any impact of dietary AhR ligands on TSLP production during papain-induced allergy. In a model of imiquimod-induced psoriasis, AhR-deficient mice had exacerbated skin inflammation (*Di Meglio et al., 2014*), while topical application of the AhR agonists FICZ (*Di Meglio et al., 2014*) or Tapinarof (*Smith et al., 2017*) ameliorated disease symptoms. Moreover, AhR-deficient keratinocytes were hyperresponsive to inflammatory stimuli ex vivo and overexpressed inflammatory mediators such as CXCL1 and CXCL5 (*Di Meglio et al., 2014*), consistent with our findings. Collectively, these observations suggest ligand-specific effects of AhR activation in keratinocytes.

In addition, intestinal type 3 ILC and lymphoid follicles are reduced in mice fed on an AhR-poor diet compared to I3C diet (*Kiss et al., 2011*), but are normal in germ-free mice (*Lee et al., 2012*), suggesting distinct effects of diet-derived and microbiota-derived AhR ligands on type 3 ILC and lymphoid follicles. The underlying mechanisms of such ligand-specific effects remain unclear (*Rothhammer and Quintana, 2019*).

In conclusion, we show that diet-derived ligands are a major source of homeostatic stimulation of AhR in keratinocytes, dampening their response to inflammation and regulating the release of bioactive Tgf-β, but not the barrier function. These results provide novel insight into the gut–skin axis and could pave the way for optimizing diet interventions to reduce the development of cutaneous allergic diseases.

## Materials and methods

### Key resources table

| Reagent type (species) or resource | Designation | Source or reference | Identifiers | Additional information |
|---|---|---|---|---|
| Strain, strain background (mouse) | *Cd207*(*Langerin*)-eGFP-DTR | *Kissenpfennig et al., 2005* | RRID:IMSR_JAX:016940 | Founder animals were donated by M.Vocanson (CIRI, Lyon, France) |
| Cell line (human) | HaCaT | *Wilson, 2014* | RRID:CVCL_0038 | |

*Continued on next page*

*Continued*

| Reagent type (species) or resource | Designation | Source or reference | Identifiers | Additional information |
|---|---|---|---|---|
| Antibody | Anti-TCRβ BUV737 (hamster monoclonal, clone H57-597) | BD Biosciences | AB_2740818 | 1:200 |
| Antibody | Anti-CD11c PerCpCy5.5 (hamster monoclonal, clone HL3) | BD Biosciences | AB_1727422 | 1:200 |
| Antibody | Anti-SiglecF BV480 (rat monoclonal, clone E50-2440) | BD Biosciences | AB_2743940 | 1:200 |
| Antibody | Anti-CD45 FITC (rat monoclonal, clone 30-F11) | BioLegend | AB_312973 | 1:200 |
| Antibody | Anti-CD11b Pe-CF594 (rat monoclonal, clone M1/70) | BD Biosciences | AB_11154422 | 1:400 |
| Antibody | Anti-MHCII BV786 (rat monoclonal, clone M5/114.15.2) | BioLegend | AB_2565977 | 1:200 |
| Antibody | Anti-Ly6G BV605 (rat monoclonal, clone 1A8) | BioLegend | AB_2565880 | 1:200 |
| Antibody | Anti-CD4 BV650 (rat monoclonal, clone RM4-5) | BioLegend | AB_2562529 | 1:200 |
| Antibody | Anti-Ly6C AF700 (rat monoclonal, clone HK1.4) | BioLegend | AB_10643270 | 1:300 |
| Antibody | Anti-EpCAM APCFire750 (rat monoclonal, clone G8.8) | BioLegend | AB_2629758 | 1:400 |
| Antibody | Anti-CD11b PerCPCy5.5 (rat monoclonal, clone M1/70) | BD Biosciences | AB_394002 | 1:400 |
| Antibody | Anti-TCRβ APC (hamster monoclonal, clone H57-597) | BioLegend | AB_313435 | 1:200 |
| Antibody | Anti-CD4 FITC (rat monoclonal, clone RM4-5) | BD Biosciences | AB_394582 | 1:200 |
| Antibody | Anti-CD19 APC-Cy7 (rat monoclonal, clone 1D3) | BD Biosciences | AB_396770 | 1:200 |
| Antibody | Anti-CD172a BUV737 (rat monoclonal, clone P84) | BD Biosciences | AB_2871154 | 1:200 |
| Antibody | Anti-CD19 BV480 (rat monoclonal, clone 1D3) | BD Biosciences | AB_2739509 | 1:200 |
| Antibody | Anti-CD3 BV480 (hamster monoclonal, clone 500A2) | BD Biosciences | AB_2744035 | 1:200 |
| Antibody | Anti-XCR1 BV510 (rat monoclonal, clone ZET) | BioLegend | AB_2565231 | 1:100 |
| Antibody | Anti-CD11c BV785 (hamster monoclonal, clone N418) | BioLegend | AB_2565268 | 1:100 |
| Antibody | Anti-CD86 FITC (rat monoclonal, clone GL1) | BD Biosciences | AB_394993 | 1:100 |
| Antibody | Anti-CD26 PE (rat monoclonal, clone H194-112) | BioLegend | AB_2293047 | 1:200 |
| Antibody | Anti-CD40 PerCP-efluor710 (rat monoclonal, clone 1C10) | eBioscience | AB_2573677 | 1:100 |
| Antibody | Anti-PDL2 APC (rat monoclonal, clone TY25) | BioLegend | AB_2566345 | 1:100 |
| Antibody | Anti-MHC II BV650 (rat monoclonal, clone M5/114.15.2) | BioLegend | AB_2565975 | 1:400 |

*Continued on next page*

*Continued*

| Reagent type (species) or resource | Designation | Source or reference | Identifiers | Additional information |
|---|---|---|---|---|
| Software, algorithm | *DESeq2* | *Love et al., 2014* | v1.22.2 | |
| Software, algorithm | GSEA | *Subramanian et al., 2005* | v4.0.3 | |
| Software, algorithm | *QuPath* | *Bankhead et al., 2017* | v.0.3.1 | |
| Software, algorithm | FlowJo | FlowJo LLC | v10 | |

## Animals

C57/B6J mice were obtained from Charles River (France). Mice were maintained on a purified diet ('AhR-poor diet,' AIN-93M, Safe diets) supplemented or not in indole-3-carbinol (I3C, 200 ppm, Sigma). In some experiments, mice were fed on a normal chow diet (4RF25 SV-PF 1609, Le comptoir des sciures). For Langerhans cells depletion, *Cd207*(*Langerin*)-eGFP-DTR mice (*Kissenpfennig et al., 2005*) were used with DTR[-/-] littermates as controls and injected intraperitoneally with 500 ng of Diphtheria Toxin (Sigma) 3 d prior to allergen treatment. For AhR in vivo inhibition, mice were treated by intraperitoneal injection of 100 µg of AhR inhibitor CH-223191 (Invivogen) on three consecutive days prior to allergen treatment. Only female mice were used, except for the Langerhans cells depletion experiments. Mice were placed on the specific diet at 5 wk of age for a period of adaptation of 3 wk, before being used for any experiment. Because mice fed on the same diet had to be housed in the same cage, no randomization was performed. Mice were maintained under specific pathogen-free conditions at the animal facility of Institut Curie in accordance with institutional guidelines. Animal care and use for this study were performed in accordance with the recommendations of the European Community (2010/63/UE) for the care and use of laboratory animals. Experimental procedures were specifically approved by the ethics committee of the Institut Curie CEEA-IC #118 (authorization APAFiS#24554-2020030818559195v1 given by National Authority) in compliance with the international guidelines. For all animal experiments, the experimental unit is a single animal. Sample size was not calculated a priori. No animal was excluded from analysis. Blinding was performed during outcome assessment.

## Allergy models

For cutaneous allergy, mice were injected in the footpad with 50 µl of phosphate buffered saline (PBS, vehicle) containing or not 50 µg of papain (Sigma). Footpad skin was collected after 6 hr or 24 hr, popliteal lymph nodes were collected after 24 hr, 48 hr, or 6 d.

For epicutaneous sensitization, mice were first shaved on the flank. The next day, 500 µg of papain, or the same volume of PBS (vehicle), was applied on a 0.5 cm$^2$ piece of sterile gauze that was then placed topically on the shaved skin. The gauze was protected by Tegaderm film (3 M), then removed after 2 hr. Inguinal lymph nodes were collected after 48 hr or 6 d.

For airway allergy, mice were anesthetized using isofluorane and exposed intranasally to 20 µl of PBS (vehicle) containing or not 10 µg of papain, at days 0, 1, 13, and 20. Blood was collected at day 20. Bronchoalveolar lavage, lungs and lymph nodes were collected at day 21.

For airway allergy after skin sensitization, mice were injected in the footpad with 50 µl of PBS (vehicle) containing or not 50 µg of papain (Sigma). At day 7, all mice were placed on the I3C diet. Mice were exposed intranasally to 20 µl of PBS (vehicle) containing or not 10 µg of papain at days 14, 15, 27, and 35. Bronchoalveolar lavage and lungs were collected at day 36.

## IgE measurement

Blood was collected and left at room temperature for 3 hr to coagulate. After centrifugation (450 × *g*, 10 min), serum was collected for analysis and kept at –20°C. IgE concentration was measured using ELISA (Invitrogen). The limit of detection was 140 pg/ml.

## Histology

Whole feet from treated mice were collected and fixed in formalin. Samples were decalcified in RDO (Eurobio Scientific) for 6 hr at 37°C before paraffin embedding. Samples were cut into 3 µm thin

sections, deparaffinized and stained with hematoxylin (Dako), eosin (RAL Diagnostics), and Safran (RAL Diagnostics) (HES). Slides were scanned with Philips ULTRA FAST scanner 1.6 RA. Images were analyzed using *QuPath* (v.0.3.1) (*Bankhead et al., 2017*).

## Electron microscopy

Ear skin was collected with a 2 µm diameter punch and was fixed in 2% glutaraldehyde in 0.1 M cacodylate buffer (pH 7.4) for 1 hr, post-fixed for 1 hr with 2% buffered osmium tetroxide, dehydrated in a graded series of ethanol solution, and then embedded in epoxy resin. Images were acquired with a digital camera Quemesa (SIS) mounted on a Tecnai Spirit transmission electron microscope (FEI Company) operated at 80kV.

## Indole measurement in mouse blood

Blood samples were collected in EDTA-coated tubes. Samples were centrifuged for 15 min at $1200 \times g$ to obtain serum. Indoles levels were measured in serum using liquid chromatography coupled to high-resolution mass spectrometry (HPLC-HRMS) (*Lefèvre et al., 2019*). Briefly, 50–100 µl of serum were added to 800 µl of methanol and samples were vortexed for 5 min in a thermomixer at 4°C followed by an incubation at −20 °C for 30 min. After centrifugation at $13,300 \times g$ for 10 min, the supernatant was collected and concentrated using a SpeedVac vacuum concentrator (Thermo Scientific). Samples were resuspended in 100 µl of 10% methanol solution, vortexed in a thermomixer for 5 min, and centrifuged at $13,300 \times g$ for 10 min. 80 µl of each sample was transferred to a liquid chromatography vial and 10 µl injected in the HPLC-HRMS system. Chromatography was carried out with a Phenomenex Kinetex 1.7 µm XB – C18 (150 mm × 2.10 mm) and 100 Å HPLC column maintained at 55 °C. The solvent system comprised mobile phase A (0.5% [vol/vol] formic acid in water), and mobile phase B (0.5% [vol/vol] formic acid in methanol). The gradient was set-up as follows: 0–2 min, 0% B; 2–7 min, 0–50% B; 7–15 min, 50–100% B; 15–18 min, 100% B, 18–18.5 min 100–0% B and 18.5–21.5 min 0%B. HRMS analyses were performed on a HPLC Vantage Flex (Thermo Fisher Scientific) coupled to a Q-Exactive Focus mass spectrometer (Thermo Fisher Scientific) that was operated in positive (ESI+). The HPLC autosampler temperature was set at 4°C. The heated electrospray ionization source was set with a spray voltage of 4.5 kV, a capillary temperature of 250°C, a heater temperature of 475°C, a sheath gas flow of 35 arbitrary units (AU), an auxiliary gas flow of 10 AU, a spare gas flow of 1 AU, and a tube lens voltage of 100 V. During the HRMS acquisition, the scan range was set to m/z = 100–500 Da, the instrument operated at 70,000 resolution (m/z = 200), with an automatic gain control (AGC) target of $1 \times 10^6$ and a maximum injection time (IT) set to automatic. Instrumental chromatography stability was evaluated by injection of a synthetic standard mixture with all metabolites of interest and quality control (QC) samples quality control obtained from a pool of the leftover of all samples analyzed. This QC sample was reinjected once at the beginning of the analysis, every 10 sample injections, and at the end of the run. Ionization and retention times were validated with pure standards and are summarized in the following table:

| Metabolite | Ion | Mass | Retention time |
|---|---|---|---|
| I3C | $[M-H_2O]^+$ | 130.0651 | 9.7 |
| L-Kynurenin | $[M+H^+]^+$ | 209.0920 | 4.1 |
| DIM | $[M-C8H7N1]^+$ | 130.0651 | 12.5 |
| I-3-AA | $[M+H^+]^+$ | 176.0706 | 8.9 |

Peak area was used as read out for relative quantification of the metabolites across the samples and normalized by serum volume.

## Skin explant analysis

Skin from the footpad was harvested 24 hr after vehicle or papain treatment. Skin explants from both footpads were pooled and incubated for 24 hr in 1 ml of RPMI (Gibco) medium containing 10% fetal calf serum (FCS, Biosera). After centrifugation ($450 \times g$, 5 min), conditioned medium was collected for analysis and kept at −20°C. TGF-β concentration was measured using ELISA (Total TGF-β1 Legend MAX and Free Active TGF-β1 Legend MAX, BioLegend). The limit of detection was 8 pg/ml for

TGF-β1. CCL2, CCL3, and CXCL1 concentration was measured using CBA (BD Biosciences). The limit of detection was 10 pg/ml.

## Flow cytometry

Cells were stained with indicated antibody cocktails supplemented with Fc block (BD Biosciences) in FACS buffer (PBS containing 0.5% BSA and 2 mM EDTA) for 30–45 min on ice. After washing with FACS buffer, cells were resuspended in FACS buffer containing DAPI (Fisher Scientific, 100 ng/ml). Cells were acquired on a ZE5 (Bio-Rad) or FACSVerse (BD Biosciences) instrument. Supervised analysis was performed using FlowJo software v10 (FlowJo LLC).

## Bronchoalveolar lavage analysis

Bronchoalveolar lavage was collected by injection of 1 ml of PBS in the bronchoalveolar space. Suspensions were filtered using 40 μm cell strainers. After centrifugation ($450 \times g$, 5 min), the lavage was collected for analysis of soluble mediators and kept at –20°C. IL5 and IL13 concentration was measured using Enhanced Sensitivity CBA (BD Biosciences). The limit of detection was 274 fg/ml.

Cells were stained with anti-TCRβ BUV737 (BD Biosciences, clone H57-597), anti-CD11c PerCpCy5.5 (BD Biosciences, clone HL3), anti-SiglecF BV480 (BD Biosciences, clone E50-2440), anti-CD45 FITC (BioLegend, clone 30-F11), anti-CD11b Pe-CF594 (BD Biosciences, clone M1/70), anti-MHCII BV786 (BioLegend, clone M5/114.15.2), anti-Ly6G BV605 (BioLegend, clone 1A8), anti-CD4 BV650 (BioLegend, clone RM4-5), anti-Ly6C AF700 (BioLegend, clone HK1.4).

## Lymph node cells analysis

For Th2 polarization analysis, lymph nodes were collected and dissociated by forcing through a 40 μm cell strainer. Cell suspensions were analyzed by flow cytometry after staining with anti-TCRβ APC (BioLegend, clone H57-597), anti-CD11b PerCpCy5.5 (BD Biosciences, clone M1/70), anti-CD4 FITC (BD Biosciences, clone RM4-5), and anti-CD19 APC-Cy7 (BD Biosciences, clone 1D3). Normalized cell numbers ($2 \times 10^5$ cells/well) were cultured for 24 hr in 100 μl of RPMI medium containing 10% FCS in the presence of 5 μl of anti-CD3/CD28 beads (Thermo Fisher). After centrifugation ($450 \times g$, 5 min), supernatant was collected for analysis and kept at –20°C. IL4, IL5, IL10, IFN-γ, IL17A, and IL13 concentration was measured using Enhanced Sensitivity CBA (BD Biosciences). The limit of detection was 274 fg/ml.

For flow cytometry of DC, lymph nodes were cut into small pieces and incubated for 30 min at 37°C in digestion mix: RPMI containing 0.5 mg/ml DNAse I (Sigma) and 0.5 mg/ml collagenase D (Roche). Cell suspensions were then filtered using 40 μm cell strainers. Antibodies used were anti-CD172a BUV737 (BD Biosciences, clone P84), anti-CD19 BV480 (BD Biosciences, clone 1D3), anti-CD3 BV480 (BD Biosciences, clone 500A2), anti-XCR1 BV510 (BioLegend, clone ZET), anti-CD11c BV785 (BioLegend, clone N418), anti-CD86 FITC (BD Biosciences, clone GL1), anti-CD26 PE (BioLegend, clone H194-112), anti-CD40 PerCP-efluor710 (eBioscience, clone 1C10), anti-PDL2 APC (BioLegend, clone TY25), anti-MHC II BV650 (BioLegend, clone M5/114.15.2), and anti-EpCAM APCFire750 (BioLegend, clone G8.8).

## Epidermal cells analysis

Skin from the footpad was harvested using scalpels. The epidermis and dermis layers were separated after incubation at 37°C for 1 hr in 0.4 mg/ml dispase II (Roche). The epidermis was collected and then cut into small pieces using scalpels and incubated for 30 min with agitation at 37°C in RPMI containing 10% FCS and 0.5 mg/ml DNAse I. Suspensions were filtered using 40 μm cell strainers.

For Langerhans cells analysis, after centrifugation ($450 \times g$, 5 min) cells were stained with anti-CD45 FITC (BioLegend, clone 30-F11), anti-EpCAM APCFire750 (BioLegend, clone G8.8) and anti-CD11b PerCPCy5.5 (BD Biosciences, clone M1/70).

For RNA-seq analysis, after centrifugation ($150 \times g$, 5 min), dead cells were removed using EasySep Dead cell removal kit (StemCell). After this step, viability was around 70% as assessed by flow cytometry. Epidermal cells were composed of 95% keratinocytes (CD45- cells) as assessed by flow cytometry.

## Imaging of epidermis

For imaging, *Langerin*-eGFP-DTR mice were used. Mice were placed on the AhR-poor or I3C diet for 3 wk. Epidermis was prepared from ear skin after hair removal using a depilating cream (Veet). The epidermis and dermis layers were separated after incubation at 4°C for 16 hr in 0.2 mg/ml dispase II (Roche). Epidermal layers were fixed in 4% paraformaldehyde for 20 min at room temperature. After washing in PBS, epidermal layers were placed on coverslips and preserved using Fluoromount-G mounting medium (SouthernBiotech).

eGFP fluorescence was imaged on an inverted laser scanning confocal microscope (Leica DMI8 with a sp8 scanning unit) equipped with an oil immersion objective (×40, NA = 1.35). The 488 nm laser was used for excitation and eGFP signal was collected on an Hybrid detector. A pixel size of 0.28 μm was chosen and Z stacks of three planes were acquired (Z step = 1 micron).

Image analysis was performed using Fiji software (*Schindelin et al., 2012*). To analyze Langerin+ cells density, a homemade macro was used. After projection of the three planes, a mask of the total eGFP was obtained. In a second step, images were blurred using a large radius (3.4 microns) to distinguish cellular stroma and count the number of cells. Finally, the number of cells was normalized to the tissue area to compute cell density.

## RNA-seq library preparation

Epidermal cells were isolated 6 hr after vehicle or papain treatment. Cells were lysed in RLT buffer (QIAGEN). Total RNA was extracted using the RNAeasy minikit (QIAGEN) including on-column DNase digestion according to the manufacturer's protocol. The integrity of the RNA was confirmed in BioAnalyzer using RNA 6000 Nano kit (Agilent Technologies) (8.8 < RIN < 10). Libraries were prepared according to Illumina's instructions accompanying the TruSeq Stranded mRNA Library Prep Kit (Illumina). 500 ng of RNA was used for each sample. Library length profiles were controlled with the LabChip GXTouchHT system (Perkin Elmer). Sequencing was performed in three sequencing unit of NovaSeq 6000 (Illumina) (100-nt-length reads, paired end) with an average depth of 40 millions of clusters per sample.

## RNA-seq data analysis

Genome assembly was based on the Genome Reference Consortium (mm10). Quality of RNA-seq data was assessed using *FastQC*. Reads were aligned to the transcriptome using *STAR* (*Dobin et al., 2013*). Differential gene expression analysis was performed using *DESeq2* (v1.22.2) (*Love et al., 2014*). Genes with low number of counts (<10) were filtered out. Differentially expressed genes between 'vehicle' and 'papain' treatment for each diet, or between 'AhR-poor diet' and 'I3C diet' conditions for each treatment, were calculated using the design 'group.' Differentially expressed genes were identified based on adjusted p-value<0.01 and Log2 FoldChange>1. Complete gene lists are included in *Figure 5—source data 1*. Heatmaps of log2-scaled expression were generated with *ComplexHeatmap*. Pathway enrichment was analyzed using *EnrichR* (*Kuleshov et al., 2016*). Sequencing data has been deposited in GEO (accession number GSE198368).

## Analysis of public transcriptomic data

Data was downloaded from GEO. Raw count matrix was normalized using *DESeq2*. For human skin explant (GSE47944) (*Di Meglio et al., 2014*), data from non-lesional skin exposed to SR1 or FICZ was used. For mouse epidermis (GSE162925) (*Uberoi et al., 2021*), data from germ-free and specific pathogen-free mice were used. Heatmaps of log2-scaled expression were generated with *ComplexHeatmap*.

## Gene set enrichment analysis

GSEA (*Subramanian et al., 2005*) was performed using the GSEA software (version 4.0.3) with the default parameters, except for the number of permutations that we fixed at n = 1000. Results are considered significant when NES > 1 and FDR < 0.25. RNA-seq data from the epidermis of germ-free and specific pathogen-free mice (GSE162925) was normalized using *DESeq2*. Gene signature for keratinocyte differentiation (GO:0045616) was downloaded from MSigDB (v7.5.1) (*Liberzon et al., 2015*). Gene signature for epidermal cells in AhR-poor diet was obtained by using the top 500 differentially

expressed genes upregulated in 'AhR-poor diet' versus 'I3C diet' for vehicle treatment, based on log2 fold change.

## Human keratinocyte culture

Human HaCaT keratinocytes were cultured with DMEM medium without glutamine or calcium (Gibco) supplemented with antibiotics (penicillin and streptomycin), 10% FCS treated with Chelex (Sigma) to removed endogenous calcium, and 0.03 mM calcium chloride (low-calcium growth medium). Cells were kept at 80% confluence in low-calcium growth medium in order to keep a basal undifferentiated phenotype (*Wilson, 2014*). For differentiation, cells were switched to the same medium containing 2.8 mM calcium chloride (high-calcium growth medium). Cells were mycoplasma-free.

HaCaT cells were exposed for 24 hr to 8 µM SR1 (Cayman Chemicals), 5 µg/ml diindolylmethane (DIM, Sigma), or 60 nM 6-formylindolo[3,2-b]carbazole (FICZ, Enzo Life Sciences) in low-calcium growth medium. Medium was then replaced by high-calcium growth medium, and cells were further cultured for 24 hr. Cells were lysed in RLT buffer and supernatants were collected for analysis using ELISA (Total TGF-β1 Legend MAX and Free Active TGF-β1 Legend MAX, BioLegend). The limit of detection was 8 pg/ml for TGF-β1.

## qPCR

Cells were harvested and lysed in RLT buffer (QIAGEN). RNA extraction was carried out using the RNAeasy micro kit (QIAGEN) according to the manufacturer's instructions. Total RNA was retro-transcribed using the superscript II polymerase (Invitrogen), in combination with random hexamers, oligo dT and dNTPs (Promega). Transcripts were quantified by real-time PCR on a 480 LightCycler instrument (Roche). Reactions were carried out in 10 µl using a master mix (Eurogentec), with the following TaqMan Assays primers (Merck): *Cyp1a1* (Mm00487218_m1), *Muc5ac* (Mm01276705_g1), *Gob5* (Mm01320697_m1), *Gapdh* (Mm99999915_g1), *B2m* (Mm00437762_m1), *Polr2a* (Mm00839502_m1), *CYP1B1* (Hs00164383_m1), *CYP1A1* (Hs01054796_g1), *ITGB8* (Hs00174456_m1), *B2M* (HS00187842_ m1), *HPRT* (Hs02800695_m1), and *RPL34* (Hs00241560_m1). The second derivative method was used to determine each Cp and the expression of genes of interest relative to the housekeeping genes (*Gapdh, B2m, Polr2a* for mouse and *B2M, HPRT, RPL34* for human) was quantified.

## Statistical analysis

Statistical tests were performed using Prism v9 (GraphPad Software). Statistical details for each experiment can be found in the corresponding figure legend. N corresponds to the number of biological replicates. Absence of asterisk indicates 'nonsignificant.' ANOVA was performed with Tukey's multiple-comparisons test.

## Acknowledgements

This work was funded by INSERM, Institut Curie, Cancéropôle Ile-de-France, Institut National du Cancer (2018-1-PLBIO-01-ICR1) and Agence Nationale de la Recherche (ANR-10-LABX-0043, ANR-10-IDEX-0001-02 PSL, ANR-17-CE15-0011-01). The authors wish to thank the Flow Cytometry Core, the Pathex Platform, the NGS Platform, the Metabolomics and Lipidomics Platform, and the In Vivo Experiments Platform of Institut Curie. The authors thank S Henri for helpful advice and M Vocanson for providing founder *Cd207(Langerin)*-eGFP-DTR mice.

## Additional information

### Funding

| Funder | Grant reference number | Author |
|---|---|---|
| Institut National de la Santé et de la Recherche Médicale | core funding | Elodie Segura |

| Funder | Grant reference number | Author |
|---|---|---|
| Institut Curie | core funding | Adeline Cros<br>Elodie Segura |
| Institut National Du Cancer | 2018-1-PLBIO-01-ICR1 | Elodie Segura |
| Agence Nationale de la Recherche | ANR-10-LABX-0043 | Mabel San Roman<br>Mathieu Maurin<br>Elodie Segura |
| Agence Nationale de la Recherche | ANR-17-CE15-0011-01 | Elodie Segura |
| Agence Nationale de la Recherche | ANR-10-IDEX-0001-02 PSL | Elodie Segura |
| European Research Council | ERC Horizon 2020-Marie Sklodowska-Curie Actions (No 842535) | Alba De Juan |

The funders had no role in study design, data collection and interpretation, or the decision to submit the work for publication.

## Author contributions

Adeline Cros, Alba De Juan, Formal analysis, Investigation, Writing – review and editing; Renaud Leclère, Sandrine Heurtebise-Chrétien, Investigation, Writing – review and editing; Julio L Sampaio, Mabel San Roman, Mathieu Maurin, Formal analysis, Investigation, Methodology, Writing – review and editing; Elodie Segura, Conceptualization, Formal analysis, Supervision, Funding acquisition, Investigation, Writing - original draft, Writing – review and editing

## Author ORCIDs

Alba De Juan http://orcid.org/0000-0003-0174-4389
Renaud Leclère http://orcid.org/0000-0001-8737-6817
Julio L Sampaio http://orcid.org/0000-0002-2881-7827
Sandrine Heurtebise-Chrétien http://orcid.org/0000-0003-1089-3829
Elodie Segura http://orcid.org/0000-0003-1795-1921

## Ethics

Animal care and use for this study were performed in accordance with the recommendations of the European Community (2010/63/UE) for the care and use of laboratory animals. Experimental procedures were specifically approved by the ethics committee of the Institut Curie CEEA-IC #118 (Authorization APAFiS#24554-2020030818559195-v1 given by National Authority) in compliance with the international guidelines.

## Decision letter and Author response

Decision letter https://doi.org/10.7554/eLife.86413.sa1
Author response https://doi.org/10.7554/eLife.86413.sa2

# Additional files

## Supplementary files

• MDAR checklist

## SeguraECrosAData availability

Sequencing data has been deposited in GEO (accession number GSE198368). Figure 5 - Source Data 1 contains the list of differentially expressed genes.

The following dataset was generated:

| Author(s) | Year | Dataset title | Dataset URL | Database and Identifier |
|---|---|---|---|---|
| Segura E, Cros A | 2023 | RNA-seq analysis of epidermal cells from mice fed with a diet deprived or not in AHR ligands | https://www.ncbi.nlm.nih.gov/geo/query/acc.cgi?acc=GSE198368 | NCBI Gene Expression Omnibus, GSE198368 |

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
