## [Editor Report]

This important study uncovers the role of Aryl Hydrocarbon Receptor (AhR) in tempering allergic responses. The authors present compelling data supporting the function of AhR ligands in limiting cutaneous allergic type 2 responses but not airway allergic responses, underscoring an interesting tissue-specific role of this axis. The work will be of broad interest to immunologists, including those with a special interest in mechanisms of regulation of allergy.

---

## [Decision Letter]

[Editors' note: this paper was reviewed by Review Commons.]

---

## [Author Response]

Additional experiments in response to Reviewer 2’s comments"The authors make a strong claim that the epidermal barrier function is not affected by AhR poor diet conditions (claim made in abstract and last paragraph of the discussion). This should be experimentally validated."

We had already performed footpad histology and observed that the stratum corneum was not affected by the diet (Figure 1A and Figure 1—figure supplement 1E). We now provide a quantitative analysis by measuring stratum corneum thickness on the images and we added this data to new supplementary figure (Figure 5—figure supplement 4). To strengthen this point, we also performed ultra-structural analysis of the epidermis using electron microscopy of the ear skin, and found no difference between diet groups (Figure 5—figure supplement 4).

"Injection into the footpad as a route of administration is also physiologically distinct from epicutaneous sensitization given the natural barriers are artificially breached via needle puncture. Did the authors consider epicutaneous sensitization via the skin without additional barrier disruption? Does this yield the same response?"

To strengthen our results, we performed epicutaneous sensitization without barrier disruption by applying papain on shaved flank skin (without any skin abrasion). We analyzed T cell responses in the draining lymph nodes after 6 days and found similar results as with footpad injection, i.e increased IL5 and IL13 secretion in mice fed on the AhR-poor diet (added as fig1E). We also analyzed dendritic cells migration to the draining lymph nodes after 48h and found that papain exposure induced cDC1 and cDC2 migration similarly in both diet groups, but Langerhans cells were reduced in mice fed on AhR-poor diet (added in Figure 4—figure supplement 1F).

Text edits in response to reviewers’ comments

Comments from Reviewer 1– " in several places they cited review articles instead of original articles for key findings. Ex. For the expression of Mucin 5 and CLCA1 a review is cited." and " the role of AHR in ILC2 (PMID: 30446384) and alveolar epithelial cells (PMID: 35935956) has been documented. The authors should add these references."

We added references to original research regarding Mucin 5 and CLCA1 : (Leverkoehne and Gruber, 2002; Nakanishi et al., 2001; Young et al., 2007). We also added references to alveolar epithelial cells in the Discussion section : "In addition, in a model of cockroach allergen-induced allergy, mice deficient for AhR in type II alveolar epithelial cells had increased airway hyperreactivity including eosinophilia and elevated Th2 cytokine secretion, due to dysregulated autophagy (Wang et al., 2022).".

Regarding ILC2, because it was shown in the mentioned article that only gut ILC2 express AhR, and in particular not lung or skin ILC2, we believe that a reference to this work is not relevant in the context of our study, and therefore we did not add this reference.

–" Although the authors mentioned treatment schedule and stimulants used in the method, a short description in the figure legend will be helpful for the readers".

We have modified all figure legends to better describe the treatment schedules.

– "1. In the introduction section, the authors should explain adequately why they thought that AHR signaling is important for allergy. "

We now better explain this point in the introduction : " In particular, AhR exerts broad functions in barrier tissues including skin and lung (Esser and Rannug, 2015), where AhR activation has been reported to limit inflammation (DiMeglio et al., 2014; Beamer and Shepherd, 2013). The impact of dietary AhR ligands in allergic responses at such barrier sites remains unknown."

Comments from Reviewer 2–"How to explain the difference between IL4 (no effect between the two diets/or absence/presence LCs in Figure 4D) and IL5/IL13 (small effect in Figure 1D and 4D). "

This is an interesting point. It has been shown that IL4 can be produced in lymph nodes by T cells distinct from those producing IL5 and IL13 (Liang et al., 2011). Consistent with this, IL4 expression is regulated in vivo by distinct mechanisms from IL5 and IL13 expression (Kim et al., 1999; Tanaka et al., 2010; Bao and Reinhardt, 2015).

We speculate that IL4-producing T cells are not affected by Langerhans cells presence. We added a point in the Discussion section to discuss this.

– "There are many more differences between germ free and specific pathogen free mice than only the presence/absence of AhR ligands. Hence, it seemed like a very big step to compare both conditions and draw the conclusion that microbiota-derived AhR ligands activate different sets of genes. It would also make more sense if Figure 5 would be immediately followed by Figure 7".

We have tuned down our conclusion regarding the different effect of diet-derived and microbiotaderived AhR ligands according to the comments of the reviewer. We now conclude: “Diet-derived AhR ligands do not affect keratinocyte barrier”. We have also moved Fig6 to the supplementary data (now Figure 5—figure supplement 3). Finally, we have modified the discussion: “In epidermal cells, we found that dietary AhR agonists regulate inflammatory pathways, but not keratinocyte barrier genes. By contrast, AhR ligands produced by skin commensal microbiota have been shown to control genes involved in keratinocyte differentiation and function (Uberoi et al., 2021). These results could be explained by ligand-specific effects.”

Other additional data

During the review of the manuscript, we had the opportunity to analyze the expression of Muc5ac in lungs in the ‘atopic march’ model. We also added a few biological replicates for this experiment. We have updated Figure 3C accordingly.

Response to other comments

Comments from Reviewer 1"In Figure 4, the authors show there is no difference in total number, but difference in migration, was there a difference in expression of migratory markers?"

We assume the reviewer is referring to the number of Langerhans cells in the epidermis in steadystate, which is not different between diets (Figure 4A). We actually already show in supplementary figure (now Figure 4—figure supplement 1E) some cell surface markers that are upregulated upon dendritic cells migration (MHC class II and CD40). We found no difference in the expression of these markers between diet groups.

“Since IL-5, IL-13 production by skin draining lymph nodes and pulmonary lymph nodes was different, is this difference due to difference in AHR expression?”

We believe that the differences in Th2 cytokine secretion in lymph nodes are due to the difference in models. In the footpad model, we analyzed an acute reaction (day 6 after papain treatment) while in the airway model, we analyzed chronic exposure (4 intra-nasal applications over 21 days).

“In Figure 3, the authors showed that intra-nasal stimulation does not induce eosinophil migration or IL-5, IL-13 induction in I3C diet group. These data and the data shown in figure-2 are in contrast. The authors should discuss this discrepancy."

In figure 3, eosinophils are actually recruited to the lungs upon papain exposure with the I3C diet (median around 5000-10000 cells in figure 3A, compared to around 100 eosinophils found with vehicle treatment in figure 2A). The reviewer’s comment shows that the representation was misleading, therefore we changed it to a log10 scale, similar to figure 2A.

Concerning cytokine detection in BAL, IL5 levels are quite similar for the I3C diet between figure 2B and figure 3B (ranging between 0 and 80 pg/mL, with a median around 20 pg/mL). Similarly for IL13, the range is quite similar between models, although the median is lower in figure 3B. This could be due to variability between experiments. Because this is a minor discrepancy, we do not believe it is necessary to add a discussion on this point in the manuscript.

Comments from Reviewer 2"Figure 1D Cytokine productionIn AhR poor diet the spread between the individual data points is much larger and the difference between presence/absence of dietary ligands in IL5 and IL13 seems to be based merely on a few outliers (which especially in the case of IL13 appear to be completely out of range). Most other datapoints do not seem to be highly different from the ones in the AhR rich diet. Where does this high variation come from in AhR poor diet (and what is the reason for these high outliers)? Would the data have been significantly different without the outliers? "

Throughout the manuscript, we have represented raw data and individual data points for transparency. We observed some variability between biological replicates, not just for cytokine secretion (fig1D) but in most assays (for instance cell counts in lymph nodes in Figure 1C or inflammatory cell counts in Figure 2A and Figure 3A or antibody production in fig2E), yet the reviewer focuses their comments on fig1D. In the case of fig1D, we have performed Kruskal-Wallis statistical tests to account for this variation, and the difference between diet groups was statistically significant. We do not understand how we could remove the so-called ‘outliers’ without data manipulation to perform an alternative statistical test. We also disagree with the reviewer that 4 out of 11 points can be considered ‘outliers’. In addition, we made similar observations in the topical application setting (added in Fig1E).

"In general, increases of all canonical T-helper cytokine responses (except for IL4) can be noted in the LN and the difference in IL10, IL17 or IFNγ production between AhR poor and rich diet appears even more pronounced than the difference in IL5/IL13 (Figure S1F). Still the authors decide to focus the entire story on the allergic response after stating that a 'lack of dietary AhR ligands amplifies allergic responses'. Why was this choice made?"

Imbalance in gut-derived AhR ligands has been shown to be involved in inflammatory bowel disease and in neuro-inflammation. The aim of the project was to address the role of dietary AhR ligands in a context that had not been previously explored. We decided to focus on allergy because AhR has broad functions in barrier tissues homeostasis, which is directly relevant to allergy. We better explained this point in the introduction: " In particular, AhR exerts broad functions in barrier tissues including skin and lung (Esser and Rannug, 2015), where AhR activation has been reported to limit inflammation (DiMeglio et al., 2014; Beamer and Shepherd, 2013). The impact of dietary AhR ligands in allergic responses at such barrier sites remains unknown."

In the course of the study, we analyzed IL10, IL17A and IFNγ production by lymph node T cells to get a complete view of helper responses, and we provided this data in supplementary information for transparency. We believe this information might be useful for other groups studying other types of skin inflammation.

"Would the authors expect other inflammatory models via the skin (e.g. bacterial, viral infection) to confer better/worse outcomes under an AhR poor diet?"

This is an interesting question. Unfortunately, we do not have the means to analyze bacterial or viral skin infections for lack of adequate facilities (i.e. BSL2 animal facility) or ethics approval for this kind of experiments. We believe that our work may prompt in the future other groups to analyze the impact of dietary AhR ligands in other inflammatory skin contexts.

"At a mechanistic level, how do LC suppress the activation of T cells in the LN, and how would this impact secretion of certain cytokines but not others?""it remains a bit speculative how migration of LCs to the dLNs of the skin contributes to suppressing Th2 immunity in the airways. Several hypotheses have been put forward in the discussion. What is their thought about this and how to validate experimentally?"

This is an important question. A regulatory role for Langerhans cells has been evidenced by other studies, but the molecular mechanisms involved remain elusive. This point is discussed in the discussion part of the manuscript. We believe that deciphering the mechanism of action of Langerhans cells is outside the scope of the present study (and is unrelated to the diet), and would represent an entire project in itself.

Figure 3 – Why would the alteration of diet pose a confounding factor to the model? Did the authors determine that such diet-associated changes are only important at the sensitization phase? The footpad (Figure 1) and pulmonary (Figure 2) models were performed with the altered diets throughout the entire length of the experiment. If anything, wouldn't changing the diet after sensitization also provide an additional variable here? Is it known what happens (e.g. inflammatory state, genetic changes) when a normal diet is resumed after a period of adaptation? This reviewer does not understand the reason for all-of-a-sudden changing the diet after the sensitization phase.

Our goal with this experiment was to address the effect of the dietary AhR ligands during the skin sensitization phase only. This is why diets are different only in this phase of the protocol. We want to emphasize that the IC3 diet and the AhR-poor diet only differ in the presence of one molecule, which is I3C. The composition of the food is otherwise exactly the same, therefore we do not believe that a change between AhR-poor and I3C would represent a confounding factor. This is different to the adaptation period when we place the mice on I3C or AhR-poor diets instead of normal chow diet (which has a completely different formulation). We made this point clearer in the text: " To this aim, we fed mice on the I3C or AhR-poor diet only during the sensitization phase, and placed all experimental groups on I3C diet 7 days after cutaneous exposure to papain or vehicle."

"Figure 7 Role of TGFbAt first site, it seems counterintuitive that TGFb, which is a molecule generally associated with homeostasis and dampening of inflammation, is associated here with more profound inflammation. How to reconcile? At this point the data on TGFb are merely correlative. Did the authors directly test the contribution of TGFb to LC migration? In addition, did they check whether they could restore defects in LC migration in absence of AhR ligands by blocking the formation of active TGFb? "

We agree with the reviewer that the role of TGFb seems counter-intuitive. However, multiple studies have shown that TGFb produced by keratinocytes retains Langerhans cells in the epidermis, using a variety of experimental approaches including genetic tools (Bobr et al., 2012; Mohammed et al., 2016; de La Cruz Diaz et al., 2021; Kel et al., 2010). We do not have any reason to doubt the validity of these studies. Therefore, we believe that demonstrating again the role of TGFb in Langerhans cells migration is not necessary.

Using blocking antibodies against TGFb or its receptor, as suggested by the reviewer, would most probably not allow us to address whether it restores the defect in Langerhans cells migration. Indeed, results from the literature (cited above) indicate that such blocking would increase Langerhans cells migration in both diet groups, therefore it will most likely be impossible to conclude.

In addition, we have provided several lines of evidence that AhR activation regulates the expression of Integrin-beta8 in keratinocytes and the release of bioactive TGFb, at transcriptomic and protein levels, in both mouse and human keratinocytes (now figure 6). Therefore, we believe that additional experiments to support the link between AhR ligands and TGFb are not necessary within the scope of the revision.

References

Bao, K., and R.L. Reinhardt. 2015. The differential expression of IL-4 and IL-13 and its impact on type-2 Immunity. Cytokine. 75:25. doi:10.1016/J.CYTO.2015.05.008.

Beamer, C.A., and D.M. Shepherd. 2013. Role of the aryl hydrocarbon receptor (AhR) in lung inflammation. Semin Immunopathol. 35:693–704. doi:10.1007/S00281-013-0391-7.

Bobr, A., B.Z. Igyarto, K.M. Haley, M.O. Li, R.A. Flavell, and D.H. Kaplan. 2012. Autocrine/paracrine TGF-β1 inhibits Langerhans cell migration. Proceedings of the National Academy of Sciences. 109:10492–10497. doi:10.1073/pnas.1119178109.

DiMeglio, P., J.H. Duarte, H. Ahlfors, N.D.L. Owens, Y. Li, F. Villanova, I. Tosi, K. Hirota, F.O. Nestle, U. Mrowietz, M.J. Gilchrist, and B. Stockinger. 2014. Activation of the aryl hydrocarbon receptor dampens the severity of inflammatory skin conditions. Immunity. 40:989–1001. doi:10.1016/j.immuni.2014.04.019.

Esser, C., and A. Rannug. 2015. The Aryl Hydrocarbon Receptor in Barrier Organ Physiology, Immunology, and Toxicology. Pharmacol Rev. 67:259–279. doi:10.1124/PR.114.009001.

Kel, J.M., M.J.H. Girard-Madoux, B. Reizis, and B.E. Clausen. 2010. TGF-β Is Required To Maintain the Pool of Immature Langerhans Cells in the Epidermis. The Journal of Immunology. 185:3248 LP – 3255. doi:10.4049/jimmunol.1000981.

Kim, J.I., I.C. Ho, M.J. Grusby, and L.H. Glimcher. 1999. The Transcription Factor c-Maf Controls the Production of Interleukin-4 but Not Other Th2 Cytokines. Immunity. 10:745–751. doi:10.1016/S1074-7613(00)80073-4.

de La Cruz Diaz, J.S., T. Hirai, B. Anh-Thu Nguyen, Y. Zenke, Y. Yang, H. Li, S. Nishimura, and D.H. Kaplan. 2021. TNF-α and IL-1β Do Not Induce Langerhans Cell Migration by Inhibiting TGFβ Activation. JID Innov. 1:100028. doi:10.1016/J.XJIDI.2021.100028.

Leverkoehne, I., and A.D. Gruber. 2002. The murine mCLCA3 (alias gob-5) protein is located in the mucin granule membranes of intestinal, respiratory, and uterine goblet cells. Journal of Histochemistry and Cytochemistry. 50:829–838. doi:10.1177/002215540205000609/ASSET/IMAGES/LARGE/10.1177_0022155402050006 09-FIGURE 2.JPEG.

Liang, H.E., R.L. Reinhardt, J.K. Bando, B.M. Sullivan, I.C. Ho, and R.M. Locksley. 2011. Divergent expression patterns of IL-4 and IL-13 define unique functions in allergic immunity. Nature Immunology 2011 13:1. 13:58–66. doi:10.1038/ni.2182.

Mohammed, J., L.K. Beura, A. Bobr, B. Astry, B. Chicoine, S.W. Kashem, N.E. Welty, B.Z. Igyártó, S. Wijeyesinghe, E.A. Thompson, C. Matte, L. Bartholin, A. Kaplan, D. Sheppard, A.G. Bridges, W.D. Shlomchik, D. Masopust, and D.H. Kaplan. 2016. Stromal cells control the epithelial residence of DCs and memory T cells by regulated activation of TGF-β. Nat Immunol. 17:414–421. doi:10.1038/ni.3396.

Nakanishi, A., S. Morita, H. Iwashita, Y. Sagiya, Y. Ashida, H. Shirafuji, Y. Fujisawa, O. Nishimura, and M. Fujino. 2001. Role of gob-5 in mucus overproduction and airway hyperresponsiveness in asthma. Proc Natl Acad Sci U S A. 98:5175. doi:10.1073/PNAS.081510898.

Tanaka, S., Y. Motomura, Y. Suzuki, R. Yagi, H. Inoue, S. Miyatake, and M. Kubo. 2010. The enhancer HS2 critically regulates GATA-3-mediated Il4 transcription in TH2 cells. Nature Immunology 2010 12:1. 12:77–85. doi:10.1038/ni.1966.

Wang, J., Y. Zhao, X. Zhang, W. Tu, R. Wan, Y. Shen, Y. Zhang, R. Trivedi, and P. Gao. 2022. Type II alveolar epithelial cell aryl hydrocarbon receptor protects against allergic airway inflammation through controlling cell autophagy. Front Immunol. 13:3879. doi:10.3389/FIMMU.2022.964575/BIBTEX.

Young, H.W.J., O.W. Williams, D. Chandra, L.K. Bellinghausen, G. Pérez, A. Suárez, M.J. Tuvim,

M.G. Roy, S.N. Alexander, S.J. Moghaddam, R. Adachi, M.R. Blackburn, B.F. Dickey, and C.M. Evans. 2007. Central Role of Muc5ac Expression in Mucous Metaplasia and Its Regulation by Conserved 5′ Elements. Am J Respir Cell Mol Biol. 37:273. doi:10.1165/RCMB.2005-0460OC.